# Toward three-dimensional hybrid inorganic/organic optoelectronics based on GaN/oCVD-PEDOT structures

Linus Krieg [1,7], Florian Meierhofer [1,7], Sascha Gorny[1], Stefan Leis[1], Daniel Splith [2], Zhipeng Zhang[2], Holger von Wenckstern[2], Marius Grundmann [2], Xiaoxue Wang[3,4], Jana Hartmann [1,5], Christoph Margenfeld[1,5], Irene Manglano Clavero[1,5], Adrian Avramescu[6], Tilman Schimpke[6], Dominik Scholz[6], Hans-Jürgen Lugauer[6], Martin Strassburg[6], Jörgen Jungclaus[1], Steffen Bornemann[1,5], Hendrik Spende [1,5], Andreas Waag [1,5], Karen K. Gleason[3] & Tobias Voss [1✉]

The combination of inorganic semiconductors with organic thin films promises new strategies for the realization of complex hybrid optoelectronic devices. Oxidative chemical vapor deposition (oCVD) of conductive polymers offers a flexible and scalable path towards high-quality three-dimensional inorganic/organic optoelectronic structures. Here, hole-conductive poly(3,4-ethylenedioxythiophene) (PEDOT) grown by oxidative chemical vapor deposition is used to fabricate transparent and conformal wrap-around p-type contacts on three-dimensional microLEDs with large aspect ratios, a yet unsolved challenge in three-dimensional gallium nitride technology. The electrical characteristics of two-dimensional reference structures confirm the quasi-metallic state of the polymer, show high rectification ratios, and exhibit excellent thermal and temporal stability. We analyze the electro-luminescence from a three-dimensional hybrid microrod/polymer LED array and demonstrate its improved optical properties compared with a purely inorganic microrod LED. The findings highlight a way towards the fabrication of hybrid three-dimensional optoelectronics on the sub-micron scale.

[1] Institute of Semiconductor Technology and Laboratory for Emerging Nanometrology, Technische Universität Braunschweig, Langer Kamp 6a/b, 38106 Braunschweig, Germany. [2] Felix-Bloch-Institut für Festkörperphysik, Universität Leipzig, Linnéstraße 5, 04103 Leipzig, Germany. [3] Department of Chemical Engineering, Massachusetts Institute of Technology, 77 Massachusetts Avenue, Cambridge, MA 02139, USA. [4] Department of Chemical and Biomolecular Engineering, Ohio State University, 151 W. Woodruff Avenue, Columbus, OH 43210, USA. [5] Epitaxy Competence Center ec2, Technische Universität Braunschweig, Hans-Sommer-Str. 66, 38106 Braunschweig, Germany. [6] OSRAM Opto Semiconductors GmbH, Leibnizstr. 4, 93055 Regensburg, Germany. [7] These authors contributed equally: Linus Krieg, Florian Meierhofer. ✉email: tobias.voss@tu-braunschweig.de

Three-dimensional (3D) semiconductor nano- and micro-wires are of substantial interest for future nanoelectronic and optoelectronic devices. GaN core-shell light-emitting diodes (LEDs) represent one prominent example[1–3], where the large emission area of a 3D nanostructure with high aspect ratio has the potential to drastically reduce the fabrication cost of white LEDs. Furthermore, the nonpolar surfaces of GaN nanowires offer optimized orientations of the material's optical dipole for efficient emission in the ultraviolet (UV) spectral range[4]. Even more, nanostructures with high aspect ratios can be completely free of extended defects, serving as perfect quasi-substrates, which might turn into a huge advantage for devices where defects play an important role in device performance, as it is the case in AlGaN UV LEDs[5,6].

All these potential advantages are counterbalanced by the challenge of chip processing in three dimensions, which is necessary to fabricate electrical contacts and optimize the overall structure in terms of photon outcoupling. At aspect ratios of above 10, many concepts of planar semiconductor technology can no longer be used.

One very interesting approach for 3D chip processing is the hybrid integration with organic functional films. Based on the combination of elasticity, flexibility, electrical conductivity, and variability in deposition methods, organic thin films should perfectly be suited for chip processing of semiconductor nanostructures with very high aspect ratios.

Oxidative chemical vapor deposition (oCVD) has been suggested to be a deposition technology that allows for the tailored, well-defined, stable, and reproducible deposition of such functional organic films. In oCVD, the synthesis of electrically conducting polymer thin films proceeds from gas-phase reactants, in contrast to conventional approaches to polymer thin film fabrication based on spin-coating from the liquid phase. In oCVD, a monomer is evaporated into a reaction chamber at medium vacuum conditions and polymerizes in a step-growth mechanism activated by a cosublimated oxidizing agent[7]. The oxidizing agent triggers the polymerization and dopes the polymer in situ. Adjusting the substrate temperature and the pressure inside the reactor gives control over properties like crystallinity or doping concentration of the resulting polymer films. Moreover, oCVD is suitable for conformal, pinhole-free coatings with homogenous thickness on all kinds of substrates including fragile or swelling materials[8] and micro- or nanostructured 3D substrates like trenches[9] or nanorods[10] where morphological characteristics are preserved. Commonly deposited polymers include poly(3,4-ethylenedioxythiophene) (PEDOT), poly(3-hexylthiophene), polyaniline, or polypyrrole[11]. In particular, oCVD PEDOT is a promising organic material for electronic applications in hybrid devices with conductivities up to 6260 S cm$^{-1}$ (ref. [12]), large tunable work functions around 5.1–5.4 eV[13] and good transparency in the visible spectral range[14–16].

So far, PEDOT, in the literature discussed as semimetallic[17], has proven its suitability in electronic applications as a p-type material. However, the use of PEDOT in its pure state without additives, as is demonstrated here, is rare. The reason can be found in PEDOT's poor solubility, which requires a mixture of PEDOT and poly(styrenesulfonate) (PSS) for the predominantly used solvent-based deposition techniques like spin-coating. Unfortunately, PSS is insulating, negatively influences the conduction paths inside the polymer film[18], and increases its resistivity. Even though the conductivity is reduced by the addition of PSS, combining PEDOT:PSS with inorganic materials results in the formation of Schottky-like electronic interfaces for PEDOT:PSS/ZnO[19,20], PEDOT:PSS/n-Si[21,22], or PEDOT:PSS/GaN[23]. However, due to the strong acidity of PEDOT:PSS, a high defect density at the hybrid interface of fragile substrates must be taken

into account[24,25]. PSS-free oCVD PEDOT overcomes these challenges.

As an inorganic counterpart for PEDOT in hybrid inorganic/organic optoelectronic structures, the well-studied direct wide-bandgap semiconductor GaN is an excellent material of choice. Several investigations highlight the application of hybrid GaN/organic structures in different fields including optoelectronics, sensing, and electronics (for a review, see ref. [26]). GaN constitutes a common platform for inorganic LEDs in everyday life and is applied in high-power electronics like high-electron-mobility transistors[27–29]. In its hexagonal wurtzite crystal structure, GaN has a direct bandgap of ~3.4 eV at room temperature[30]. In combination with AlN and InN, GaN forms—in theory—a continuous alloy system whose bandgap ranges from 0.7 eV (pure InN)[31,32] to 6.2 eV (pure AlN)[30].

Here, we address two key challenges on the way to 3D hybrid optoelectronics: first, the fabrication of the organic–inorganic hybrid structures, and second, the control and analysis of the electrical properties of the hybrid interfaces. We demonstrate the potential of oCVD as a deposition technology to conformally coat GaN-microrod structures with pure PEDOT. In the next step, we study the electronic properties of planar hybrid structures consisting of conformal oCVD PEDOT layers fabricated directly on an n-GaN substrate for optoelectronic applications. Temperature-dependent current–voltage (I–V) measurements reveal pronounced diode-like characteristics with rectification ratios of about 10$^7$ at ±2 V with high temporal, thermal, and electrical stability. Modeling the I–V characteristics based on a Schottky model allows for the deduction of barrier heights, ideality factors and the development of a band-alignment model. The deduced ideality factors strengthen the claim of a low defect density at the hybrid interface due to the gentle deposition technique. Ideality factors and effective barrier heights are consistent with the model of potential fluctuations at the interface[33] supporting the concepts of thermionic emission as the predominant conduction mechanism across the hybrid interface and oCVD PEDOT as having metallic properties. The capability of oCVD PEDOT in hybrid inorganic–organic LEDs is discussed in detail for a planar structure in which p-type oCVD PEDOT replaces the commonly used p-GaN layer. GaN/PEDOT hybrid LEDs show a bright and homogenous light emission with an enlarged light-emitting area as compared to a conventional, inorganic reference LED at similar applied voltages and contact sizes. This enlargement is attributed to the improved lateral current spreading as compared to p-GaN. Finally, as a proof of principle, we report electroluminescence (EL) from a hybrid n-GaN/oCVD-PEDOT microrod array and compare its improved optical properties with a purely inorganic GaN-based microrod LED array.

## Results

**3D n-GaN/PEDOT structures.** Conformal and homogenous coatings of microstructured substrates demand high precision and good control over the applied deposition technique for the coating. Figure 1 and Supplementary Fig. 1 demonstrate the advantage of oCVD as compared to solvent-based spin-coating for the fabrication of core-shell microrod structures. In Fig. 1a, uncoated metal-organic vapor phase epitaxy (MOVPE) grown GaN-microrods are shown. The microrods have a well-defined hexagonal shape with six m-plane side facets, a diameter around 700 nm, a length of about 3 μm, and a pitch of 2.4 μm. Coating the microrods with PEDOT using oCVD (Fig. 1b) preserves their morphological characteristics.

From scanning electron microscopy (SEM) images, the PEDOT film thickness was estimated to be around 60 nm. The main difference between uncoated and coated microrods can be

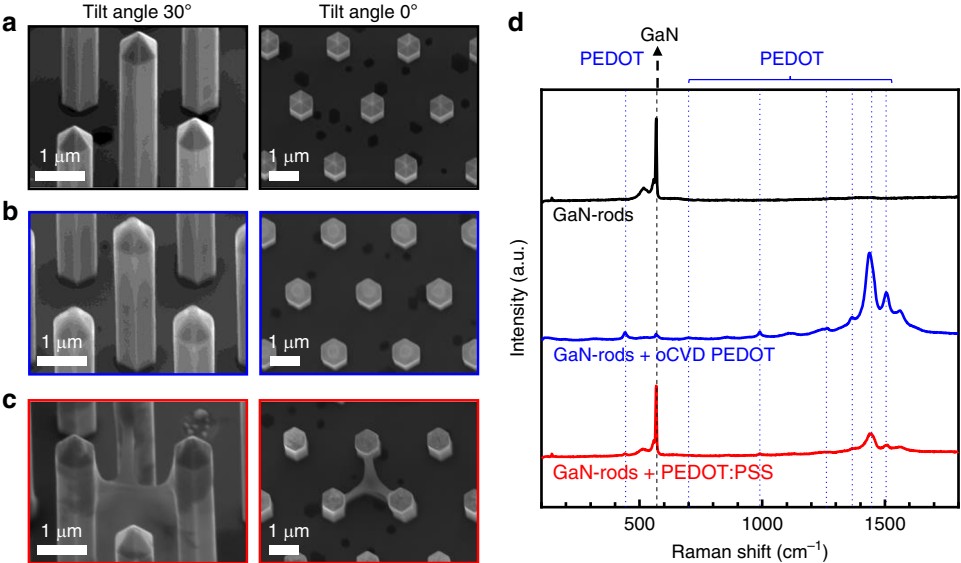

**Fig. 1 SEM micrographs and Raman signals of uncoated and coated GaN-microrods. a** Uncoated and **b** conformally coated microrods with organic PEDOT deposited via oxidative chemical vapor deposition (oCVD). **c** Nonconformal coverage of GaN-microrods with PEDOT:PSS via spin-coating in which surface tension effects lead to liquid bridging between three adjacent microrods (secondary electron map, tilt = 30°/0°, $E_{PE} = 5$ kV, scale bars correspond to 1 μm). **d** Corresponding Raman signals reveal the characteristic GaN peak (dashed, black) and PEDOT peaks (dotted blue) for the coated microrods.

seen at the edges of the microrods which are slightly smoother in the coated case. Coating the microrods with PEDOT:PSS using the well-established spin-coating technique (Fig. 1c), however, leads to substantially less homogenous polymer layers or even bridges of the material spin-coated between the microrods. Raman spectroscopy (Fig. 1d) confirms the presence of PEDOT layers as the actual coatings of the microrods. The uncoated microrods clearly show the GaN peak at 567 cm$^{-1}$ corresponding to the $E_2$ phonon frequency of unstrained GaN[34,35]. This peak exists also for the PEDOT:PSS coated microrods and, to a minor extent, for those coated with oCVD PEDOT. In addition, the oCVD-coated microrod structures feature the characteristic Raman signals for PEDOT which are attributed to asymmetric $C_\alpha = C_\beta$ stretch (1505 cm$^{-1}$), symmetric $C_\alpha = C_\beta$(-O) stretch (1436 cm$^{-1}$), $C_\beta$-$C_\beta$ stretch (1365 cm$^{-1}$), $C_\alpha$-$C_{\alpha'}$ interring stretch (1265 cm$^{-1}$), C-O-C deformation (1105 cm$^{-1}$), oxyethylene ring deformation (991 cm$^{-1}$), and symmetric C-S-C deformation (700 cm$^{-1}$)[36,37]. The GaN-microrods spin-coated with PEDOT:PSS have less pronounced PEDOT lines which could be ascribed to a less coherent, incomplete polymer thin film. Therefore, oCVD has proven its suitability for achieving homogeneous coatings of 3D GaN-based microrods, where it can be considered as the deposition method of choice for conductive polymers.

**Electrical characterization of planar n-GaN/PEDOT structure.** The electronic properties of hybrid n-GaN/PEDOT interfaces are studied on planar reference structures. Figure 2a shows the sketch of our planar GaN/PEDOT device, while Fig. 2b displays a photograph of the completed device. Two additional samples (a nominally identical reference sample and a sample with a different doping concentration of the GaN layer) are discussed in Supplementary Fig. 2 and Supplementary Note 1. The structure consists of a 4 μm thick Si-doped n-GaN layer grown by MOVPE on a c-plane sapphire substrate (not shown). Electrochemical capacitance-voltage (ECV, Supplementary Fig. 3) measurements yield a donor density of $2 \times 10^{18}$ cm$^{-3}$, comparable to the n-GaN doping density used in LED structures[38,39]. One half of the GaN layer is covered with an array of PEDOT squares of thickness 110 nm, which square dimensions range between 100 and 400 μm on

a side. As contacts, a 100 nm thick gold film was evaporated onto the PEDOT squares, and the contacts on n-GaN consist of a stack of 100 nm gold on 30 nm titanium. Supplementary Fig. 4 shows SEM images of the contacts of an equivalent sample highlighting the well-defined contacts and the PEDOT beneath. The roughness of the PEDOT layer is estimated to be in the range of 1–10 nm as reported by earlier studies of oCVD PEDOT layers[12,40,41] and an AFM measurement on an oCVD PEDOT layer deposited in the same reactor at comparable growth conditions (Supplementary Fig. 4f). Details of the sample fabrication can be found in the "Methods" Section, Supplementary Methods and were reported elsewhere[12,37].

*I–V* measurements reveal a clear diode-like behavior which is fully attributed to the hybrid n-GaN/PEDOT heterojunction. Figure 2c shows the initial *I–V* measurement 2 weeks after the deposition of the oCVD PEDOT layer as well as a follow-up test after 113 weeks of aging at ambient conditions in the dark and the characteristics after heating the sample to 423 K in air and cooling back to room temperature. The electrical properties of the hybrid structure show excellent temporal and thermal stability. The high rectification ratio of $10^7$ at ±2 V remains stable over a period of more than 2 years and indicates excellent temporal stability under ambient conditions. The leakage current density is about 1 μA cm$^{-2}$ up to −4 V. The excellent temporal stability is impressive as a degradation of the PEDOT layer was observed in earlier studies. Chen et al.[42] investigated the degradation of oCVD PEDOT and distinguished between two mechanisms of degradation, both based on diffusion processes: on the one hand, diffusion of $H_2O$ and $O_2$ into the film and on the other hand diffusion of the dopant out of the film. However, the degradation of the conductivity in out-of-plane direction was not investigated. Due to the Au encapsulation on top of our polymer layer, we could neither observe nor measure any degradation effects.

Temperature-dependent *I–V* measurements were performed during the initial measurement period 2 weeks after the deposition of the polymer layers. The *I–V* characteristics slightly change after a heat cycle up to 423 K. The rectification ratio at ±2 V is one order of magnitude smaller after heating. The reduction is caused

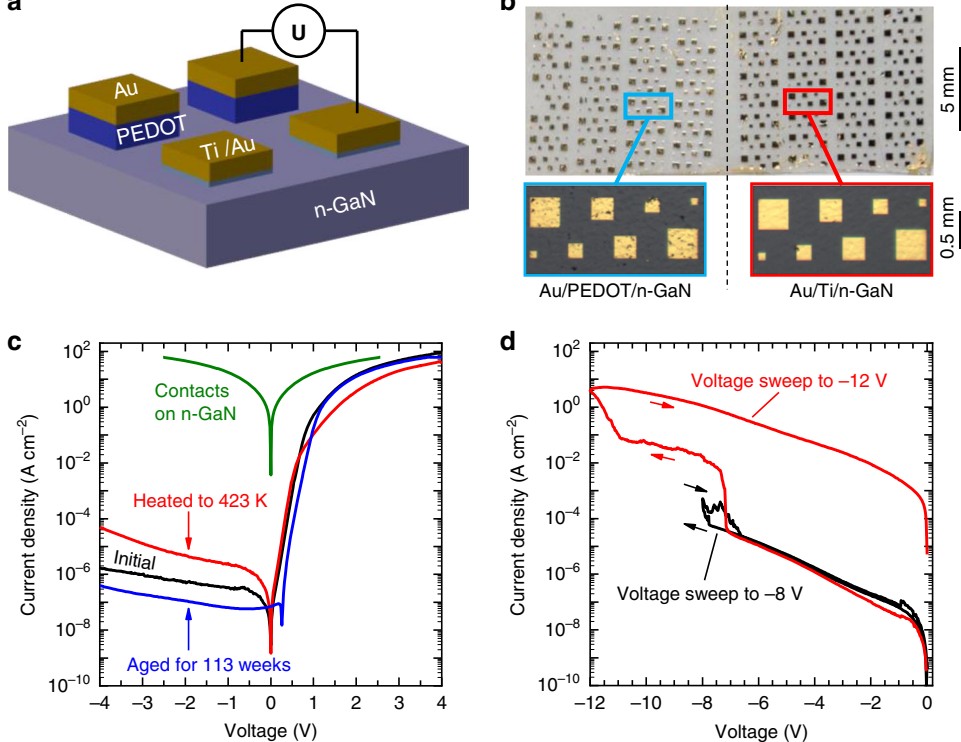

**Fig. 2 Design and electrical characterization of planar heterojunction structure. a** Sketch of the sample. **b** Photography (scale bar: 5 mm) of as-prepared devices with magnified view (scale bar: 0.5 mm) of the p- and n-type contact pads, respectively. **c** I–V characteristics of the hybrid n-GaN/PEDOT heterojunctions. Minor differences from the original performance (black) occur after either heating to 423 K and cooling back to 293 K (red) or 113 weeks of aging at ambient conditions in the dark (blue). The overall forward current density is limited by the series resistance of n-GaN as determined by I–V measurements of Au/Ti/n-GaN/Ti/Au structures (green). **d** Determination of the breakdown voltage, soft (black) and hard (red) breakdown occur at −7.8 and −10.8 V, respectively.

on the one hand by an increase of the leakage current by one order of magnitude and a decrease of the on-current by a factor of four on the other hand. Also, the shape of the I–V characteristic for negative voltages changed slightly. Overall, good rectifying properties persist. To get an overview of the electrical characteristics during the heating and cooling process, all obtained characteristics for heating and cooling are shown in Supplementary Fig. 5. The breakdown voltage of the hybrid structure is estimated to be about −7.8 V (Fig. 2d and Supplementary Note 2). This breakdown was reversible, and the characteristics returned to previously recorded values when applying reverse voltages smaller than −6.5 V. At a voltage of around −10.8 V, a hard irreversible breakdown occurs. The rectification of the device is lost for all following measurements indicating the formation of ohmic channels or shunts through the device.

Reference measurements on test structures (Au/PEDOT/Au) confirmed linear I–V curves (Supplementary Fig. 6). Au/Ti/n-GaN/Ti/Au showed deviations from perfect linear behavior for voltages between −0.5 and 0.5 V (green curve in Fig. 2c and Supplementary Fig. 6c). However, these deviations alone are not sufficient to explain the large rectification ratio that is experimentally found for the hybrid diode as the magnitude of the current density of the contacts is much larger than the diode contribution. We note that the current density in forward direction of the hybrid device approaches the values obtained in the contact measurements (Au/Ti/n-GaN/Ti/Au). Therefore, we conclude, that the current density for voltages larger than 4 V is limited by the series resistance of the n-GaN layer rather than that of the PEDOT layer. A more detailed discussion of the contacts can be found in Supplementary Note 2.

**Schottky model for hybrid n-GaN/p-PEDOT structures.** Given the excellent I–V characteristics of the hybrid n-GaN/p-PEDOT structures discussed in the previous section, we can apply fundamental models to define and extract characteristic parameters like electronic barrier heights and ideality factors. This way, we can compare the performance of different devices and also develop strategies for optimization. Hybrid inorganic–organic diodes consisting of conductive organic layers on GaN are commonly analyzed in the frame of a Schottky model[23,26,43,44]. The Schottky model refers to a metal–semiconductor contact where an electronic barrier forms at the interface[45]. This is also the model of choice for the data analyzed in this paper. Several considerations justify this approach. First, Wang et al.[12] reported the metallic behavior of oCVD PEDOT. In combination with the semiconducting material n-GaN, we have, thus, a metal–semiconductor interface. Secondly, the current density of the hybrid device is limited by the conductivity of the n-GaN layer rather than that of the PEDOT, as discussed in the previous section. Therefore, we assume a one-sided junction and thermionic emission as the dominant conduction mechanism.

In the frame of the Schottky model, the current I can be written as

$$I = I_s \left( \exp\left( \frac{e(U - R_s I)}{k_B T n} \right) - 1 \right) + \frac{U - R_s I}{R_p}, \quad (1)$$

where e is the elementary charge, U the applied voltage, $R_s$ the series resistance of the structure, $R_p$ the parallel resistance, $k_B$ the Boltzmann constant, and T the temperature at which the measurement was performed. n is the ideality factor which is a measure for the bias dependence of the barrier height. For ideal Schottky diodes, no bias dependence of the barrier height is

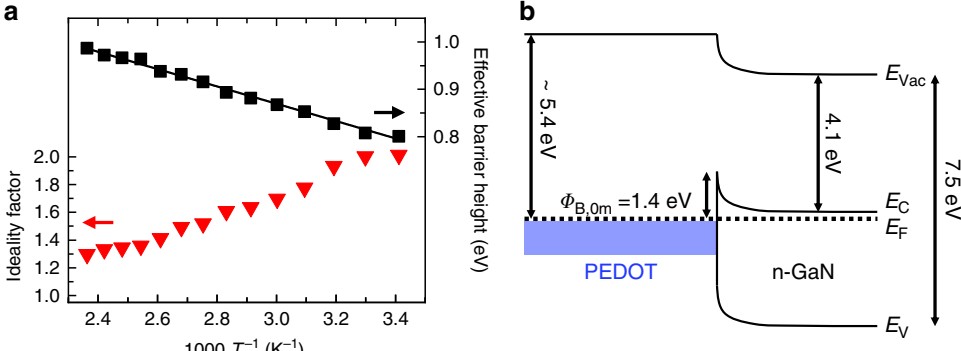

**Fig. 3 Diode characteristics obtained for the hybrid GaN/PEDOT specimen. a** Ideality factors and effective barrier heights obtained from the thermionic emission model for *I–V* measurements on the hybrid GaN/PEDOT specimen show temperature dependencies which are characteristic for lateral inhomogeneities. Evaluation of the characteristics with the model of thermionic emission over a laterally inhomogeneous barrier yields a mean barrier height of about $\Phi_{B,0m} = 1.4$ eV. The solid line represents a linear fit according to Eq. (3) for the effective barrier height. **b** Band diagram of the hybrid n-GaN/PEDOT structure. $E_V$, $E_F$, $E_C$, and $E_{Vac}$ correspond to the valence band, Fermi level, conduction band and vacuum level, respectively.

expected, leading to an ideality factor of 1. Values of $n > 1$ were discussed to be related to lateral inhomogeneities of the barrier, interface layers, high doping levels, large fields or traps in the space-charge region (for a short discussion see, i.e., Werner and Güttler[33]). $I_s$ is the saturation current which can be expressed as

$$I_s = A_0 A\, T^2 \exp\left(\frac{-\phi_{B,eff}}{k_B T}\right). \qquad (2)$$

It comprises the effective Richardson constant $A$ ($A = 26.64$ A K$^{-2}$ cm$^{-2}$ for n-GaN with an effective electron mass of $0.222 m_0$)[46], the contact area $A_0$, the temperature $T$, elementary charge $e$, and the effective Schottky barrier height of the hybrid interface at zero bias $\Phi_{B,eff}$. The first term of Eq. (1) describes the actual diode characteristics, the second term accounts for parallel electrical channels, such as shunts. In Supplementary Fig. 7, we depict the schematic equivalent circuit.

Figure 3a shows the ideality factors and effective barrier heights obtained by fitting Eqs. (1) and (2) to our experimental data for a temperature series, where the sample is heated from room temperature to 423 K. Supplementary Fig. 8 gives, in addition, the values determined from cooling the sample back to room temperature. The model yields ideality factors ranging from $2.01 \pm 0.05$ at room temperature down to $1.30 \pm 0.05$ at 423 K. The effective barrier height increases significantly from 0.80 to 0.99 eV, respectively. The values obtained during the heating and cooling process of the sample at the same temperatures are consistent and support the hypothesis of the good thermal stability of the hybrid junctions (Supplementary Fig. 8).

All values of the ideality factors are among the lowest reported values compared to those of PEDOT:PSS deposited on n-GaN via spin-coating that range from 1.3 to 12.9[23,26,43,44]. They signify the high quality of the hybrid interface with a low defect density and emphasize the advantage of oCVD as being a suitable deposition technique for the fabrication of high-quality hybrid inorganic/ organic interfaces.

The clear variation of the ideality factor and barrier height with the sample temperature is explained by Werner and Güttler in the frame of a model based on thermionic emission that additionally considers fluctuations of the local electronic potential at the hybrid interface[33]. The main idea is that ideality factors $n > 1$ are the result of lateral inhomogeneities at the metal/semiconductor interface leading to a deformation of the spatial barrier distribution under applied bias voltages. These fluctuations of the barrier height are modeled by a Gaussian distribution with a mean barrier height at zero bias $\Phi_{B0,m}$ and a standard deviation $\sigma$. According to the model, the effective barrier height $\Phi_{B,eff}$, as

obtained from *I–V* measurements, can be described as

$$\phi_{B,eff}(T) = \phi_{B0,m} - \frac{\sigma^2}{2 k_B T}. \qquad (3)$$

Fitting Eq. (3) to the effective barrier height of our sample (Fig. 3a) yields a mean barrier height at zero bias of $\Phi_{B0,m} = (1.42 \pm 0.01)$ eV with a standard deviation $\sigma = (218 \pm 10)$ meV. The mean barrier height is in good agreement with the barrier obtained by capacitance–voltage (*C–V*) measurements at room temperature, which yield a value for the electronic barrier of $\Phi_{B,CV} = (1.39 \pm 0.01)$ eV. As the potential fluctuations of the potential barrier are small compared to the depth of the depletion region, *C–V* measurements do not take into account potential fluctuations at the interface and are therefore comparable with the mean barrier height[47]. Further evaluations within the model of Werner et al. and a complementary, more empirical approach to determine homogenous barrier heights from *I–V* measurements based on a model of Schmitsdorf et al.[48] are presented in Supplementary Notes 3 and 4 and Supplementary Fig. 9.

Taking the mean barrier height at zero bias as obtained by the model of Werner and Güttler, we can deduce a band alignment diagram for the hybrid interface in the frame of a Schottky model (Fig. 3b). All other values are taken from literature: the work function of oCVD PEDOT is reported to be ~5.4 eV and varies with the temperature of the substrate during the oCVD process[12]. GaN has an electron affinity of 4.1 eV and a bandgap of 3.4 eV[30]. We note that the obtained effective barrier height of 1.42 eV matches well the difference between the work function of PEDOT and the electron affinity of GaN. This is expected for the case of an ideal Schottky barrier if surface effects like surface states do not play a role[45].

The discussion of the *I–V* characteristics based on the model of potential fluctuation within the thermionic emission model by Werner and Güttler yields consistent results. Hence, thermionic emission can be taken as the dominant conduction mechanism at the hybrid interface. In addition, as the model is based on a metal–semiconductor Schottky diode, the assumption of PEDOT as having metallic characteristics is strengthened. The obtained mean barrier height matches the difference between the assumed work function of PEDOT and the electron affinity of GaN. Ideality factors close to the ideal values emphasize the high quality of the hybrid interface and hereby a low trap density thus implying negligible trap-assisted recombination. Furthermore, the values between 2 and 1.4 indicate that the deviation from the ideal *I–V* characteristics is not as pronounced as for other n-GaN/ PEDOT:PSS structures reported in the literature. A higher

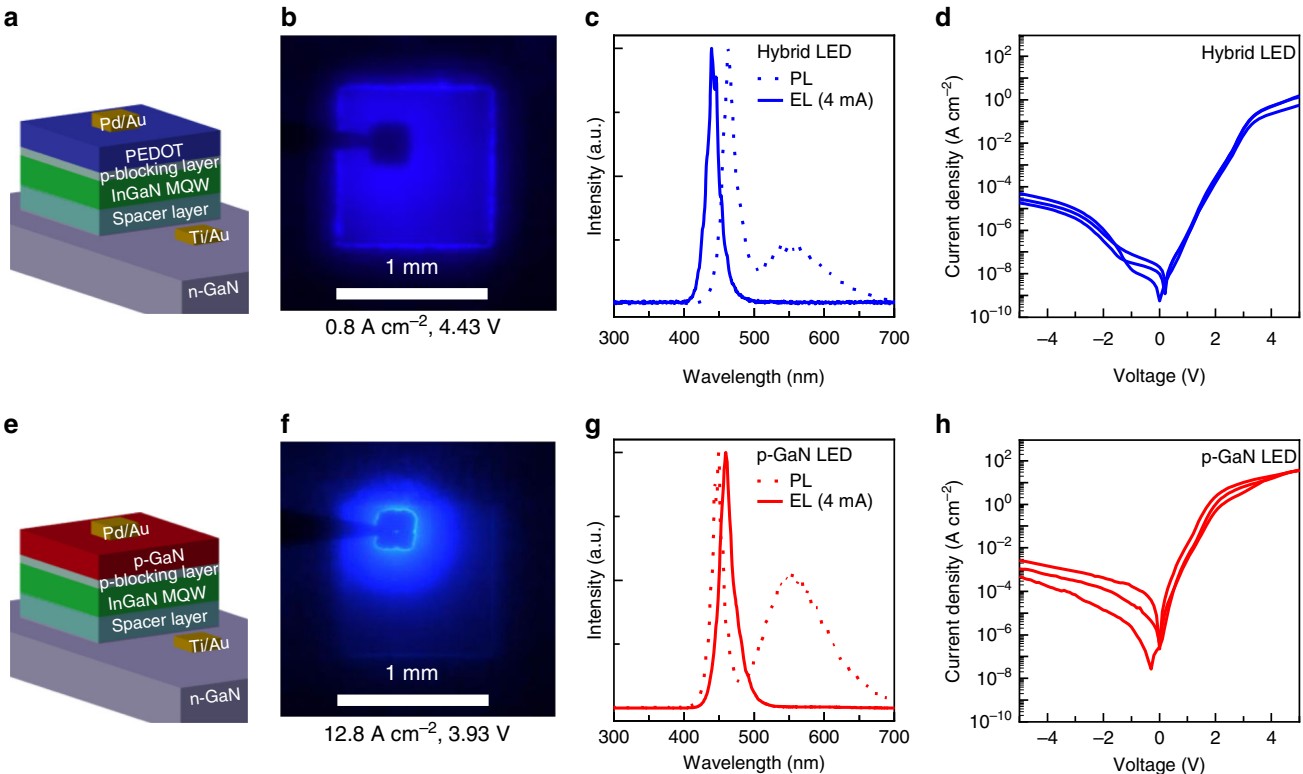

**Fig. 4 Comparison of a hybrid and a conventional LED structure. a** Sketch of the hybrid inorganic–organic LED where the p-GaN top layer was replaced with PEDOT. **b** Optical image of the hybrid LED at an applied voltage of 4.43 V reveals electroluminescence (EL) for the entire 1 × 1 mm$^2$ active area since PEDOT serves as a current spreading and hole-injection layer (scale bar: 1 mm). **c** Spectral analysis of photoluminescence (PL) excited at 325 nm and EL identifies the InGaN multi-quantum well as origin of the luminescence. **d** I–V measurements obtained for three identically constructed hybrid devices yield a series resistance of (170 ± 79) Ω. **e** Sketch of the conventional, purely inorganic p-GaN-based LED. **f** The optical image of the conventional planar LED at an applied voltage of 3.93 V features a smaller light-emitting area of the EL due to the limited lateral conductivity of the p-GaN layer (scale bar: 1 mm). **g** PL and EL peak wavelengths at ~450 nm confirm that the multi-quantum wells of the conventional and hybrid LEDs are similarly designed. **h** I–V analysis of the p-GaN-based device reveals a series resistance of (84 ± 7) Ω.

crystallinity of the PEDOT layers that can be achieved with higher substrate temperatures[12] or liquid oxidants[15] during the deposition process of the organic layer would most likely further improve the electrical characteristics. The high quality and reproducibility of the Schottky barrier formation at the hybrid interfaces implies the suitability of these structures for electronic and optoelectronic applications.

**Planar hybrid LED structure**. The structures analyzed so far show diode characteristics with high rectification ratios. However, no EL could be detected in the tested voltage range. This changes if we insert an active layer between n-GaN and p-PEDOT. To realize such a hybrid inorganic/organic LED, we use a conventional inorganic GaN-based LED structure without the terminating p-GaN layer. The LED structure consists of an n-GaN substrate, an InGaN multi-quantum well (MQW) and a thin p-AlGaN electron blocking layer (Fig. 4a). The electron blocking layer balances the mobilities of electron and hole charge carriers and leads to a more efficient carrier capture of electrons into the quantum well. The MOVPE growth of the structure is terminated after the p-AlGaN electron blocking layer and an oCVD PEDOT layer with a thickness of about 640 nm is deposited instead of the commonly used p-GaN.

Applying a constant current of 8 mA to the structure results in intense blueish EL that can easily be detected with the bare eye or an optical microscope (Fig. 4b). The darker area in the center of the micrograph is the contact pad with the contact needle on the

left, which is blocking the emitted light. The homogeneous bright blue emission centering at a wavelength of 440 nm originates from the whole 1 × 1 mm$^2$ rectangular LED structure and can be attributed to the recombination of electron-hole pairs in the InGaN-MQW as determined by the comparison of EL and photoluminescence (PL) measurements (Fig. 4c). The spectral shift between the InGaN-emission of the PL (440 nm) and EL (462 nm) measurements is attributed to fluctuations of the indium concentration within the structure as the measurement position on the sample is different. The broad PL emission around 560 nm is attributed to GaN defect states. I–V characteristics of three identically designed LEDs (Fig. 4d) suggest good sample reproducibility and indicate a rectification ratio in the order of 10$^4$ at ±4 V.

Taking a conventional inorganic LED structure with the terminating p-GaN layer as a reference (Fig. 4e) we also observe EL (Fig. 4f). Once again, the PL maximum at ~450 nm and EL maximum at ~460 nm (Fig. 4g) help to identify the origin of the luminescence as recombination in the InGaN-MQW. The I–V characteristics of the conventional LED structure (Fig. 4h) also result in a rectification ratio in the order of 10$^4$ at ±4 V. The series resistance dominates the conventional LED structure at about 2 V, the hybrid structure is limited by the series resistance above 3.5 V.

In comparison to the hybrid LED structure, the defect luminescence around 560 nm seems to be more pronounced in the conventional LED. The area of light emission in the conventional LED is restricted to the area beneath the contact

pad and no further visible lateral current spreading takes place. The reason lies in the low lateral conductivity of the p-GaN top layer in the conventional LED. The lateral conductivity of the PEDOT layer as estimated by a PEDOT/glass sample from the same oCVD run like the hybrid LED is $(75 \pm 15)$ S cm$^{-1}$. Calculations for bulk p-GaN based on literature values (hole mobilities up to 20 cm$^2$ V$^{-1}$ s$^{-1}$ and hole concentrations up to $10^{18}$ cm$^{-3}$)[30] yield a value of about 3.2 S cm$^{-1}$ which is about 25 times smaller than the value for PEDOT. This emphasizes the importance of the PEDOT layer with its substantially larger lateral conductivity and suitable energy levels for efficient hole injection for large area EL from the full hybrid LED. Due to the larger light-emitting area of the hybrid device, the estimated current density is much lower as compared to the conventional LED where the active light-emitting area is much smaller (~0.8 and ~12.8 A cm$^{-2}$, respectively). This explains the brighter EL of the conventional LED. We note that for an applied current of 8 mA the obtained voltages are in the same order with 4.43 and 3.93 V for the hybrid and conventional LED structure, respectively.

The observations therefore clearly demonstrate that the PEDOT layer is an interesting alternative to transparent conductive oxides or even to p-GaN in GaN LEDs. The oCVD process enables the fabrication of pure PEDOT layers, which leads to conductivities far beyond the ones for p-GaN. Holes could be injected into the quantum wells under an applied bias from the hybrid p-side, even though a Schottky barrier of 1.42 eV has been identified for the hybrid GaN/PEDOT interface.

**3D Microrod hybrid LED structure.** The increased lateral conductivity of the PEDOT layer is particularly crucial for 3D LED structures to profit from the increased surface to volume ratio of the structures without shielding the light emission. Putting the results together allows for a straightforward proof of principle approach to realize a 3D hybrid LED structure. Two core-shell systems were designed that differ in the type of the outer p-type shell layer, only (schematic cross-sectional shell designs shown in Fig. 5a, b). As described in Supplementary Methods, they were fabricated in a selective area growth MOVPE process using a SiO$_x$ mask. Around the n-doped GaN core, an InGaN-MQW is grown and capped with a thin p-doped AlGaN electron-blocking layer. Whilst the conventional purely inorganic reference core-shell microrod structure is terminated with a MOVPE-grown p-GaN shell (Fig. 5b), the microrod array for the hybrid structure is coated with oCVD PEDOT (Fig. 5a). Both samples are patterned into several squares of microrod arrays (~75 μm × ~75 μm) using laser ablation. SEM, PL, and Raman spectroscopy maps clearly reveal that the laser ablation process removed all microrods outside the array and identify a PEDOT coating just on the microrod array including the underlying substrate (Supplementary Fig. 10). An average thickness of $(86 \pm 14)$ nm and $(45 \pm 14)$ nm is obtained for the PEDOT and p-GaN shell, respectively, by comparing rods with and without p-type shell (Supplementary Fig. 11).

To perform cathodoluminescence (CL) and EL measurements on a single microrod, the samples were transferred into an SEM chamber equipped with micromanipulators and a CL system for light collection. Figure 5c, d shows SEM images (left) of the individual LED rods and their corresponding hyperspectral CL images (right) which were obtained by recording 30 individual monochromatic CL images ($\lambda_{\text{m-CL}}$) in the spectral range between 390 nm and 450 nm. For both structures, the color-coded wavelength of the intensity maximum of the CL emission clearly demonstrates a red shift when going from the bottom to the top of the microrods. The red shift of the main CL peaks as obtained

from CL spectra (Fig. 5e, f) comprises 409 to 447 nm and 414 to 462 nm for the hybrid and the purely inorganic microrods, respectively. The shift is attributed to a gradient in the indium concentration as additionally confirmed by PL scanning along the microrod height (Supplementary Fig. 12) and reported in the literature[49,50].

The indium gradient along the height of the microrod is also reflected in the EL measurements in Fig. 5g, h. The n-contact for the EL measurements is realized by contacting the n-doped GaN buffer next to the microrod array. For the p-contact, one micromanipulator is connected to the p-type shell of a microrod (insets of Fig. 5g, h). Figure 5g displays the EL detected for the hybrid LED structure. The luminescence has a broad distribution with two peaks at 415 and 475 nm. These two peaks reasonably match the CL (Fig. 5e) and PL (Supplementary Fig. 12e) signals corresponding to the indium fluctuations of the MQW along the height of the microrod. In contrast to the hybrid structure, the p-GaN-based LED in Fig. 5h has clearly defined EL maxima at 461 and 430 nm depending on the position of the contact needle at the top or the bottom of the microrod, respectively. The coexistence of both maxima in the EL spectrum of the hybrid microrod LED demonstrates the enhanced current spreading of the PEDOT layer compared to the p-GaN layer. The PEDOT layer allows, therefore, to take advantage of the full height of the core-shell microrod as the light-emitting region of the LED. As the PEDOT contact layer additionally covers the substrate between the individual microrods, we expect the current spreading to even reach the neighboring microrods, hence leading to a significant difference in the effective current density as compared to the purely inorganic 3D LED.

## Discussion

Hybrid optoelectronic devices based on core-shell microrod structures require conformal and pinhole-free coatings that can barely be accomplished by conventional liquid-based deposition techniques. P-conductive polymer layers deposited by oCVD meet these requirements. In particular, the demonstration of EL from 3D hybrid inorganic–organic LEDs proves the suitability of our approach for the fabrication of complex hybrid devices for optoelectronics. $I$–$V$ measurements of planar n-GaN/oCVD-PEDOT reference structures verify the formation of Schottky diodes with rectification ratios of up to $10^7$ at $\pm 2$ V and demonstrate excellent thermal and temporal stability, another prerequisite for reliable applications in optoelectronics. Ideality factors and effective barrier heights obtained for the hybrid interface are consistent, among the lowest reported values for comparable n-GaN/PEDOT structures and strengthen the idea of thermionic emission as the dominant conduction mechanism at the hybrid interface. Inserting an InGaN multi quantum-well into the hybrid structure results in pronounced EL with the hybrid structure featuring a substantially enlarged light-emitting area due to an improved current spreading as compared to conventional fully inorganic GaN-based LED structures.

In summary, with the strength of oCVD that offers controllable and homogeneous coatings on planar and 3D structures and the diode formation of n-GaN/PEDOT structures, the realization of complex 3D microstructures for optoelectronic devices is in reach. From the findings of this paper, we conclude that oCVD PEDOT can be a key component to realize hybrid inorganic/organic 3D LEDs based on GaN microstructures.

## Methods

**Oxidative chemical vapor deposition.** For the oCVD growth process, the substrates are mounted inside the vacuum chamber (~$10^{-4}$ mbar, unless differently stated) top-down on a heatable stage ($T_{\text{substrate}}$) (Supplementary Fig. 13). Simultaneously, EDOT (3,4-ethylenedioxythiophene, purchased from Sigma Aldrich)

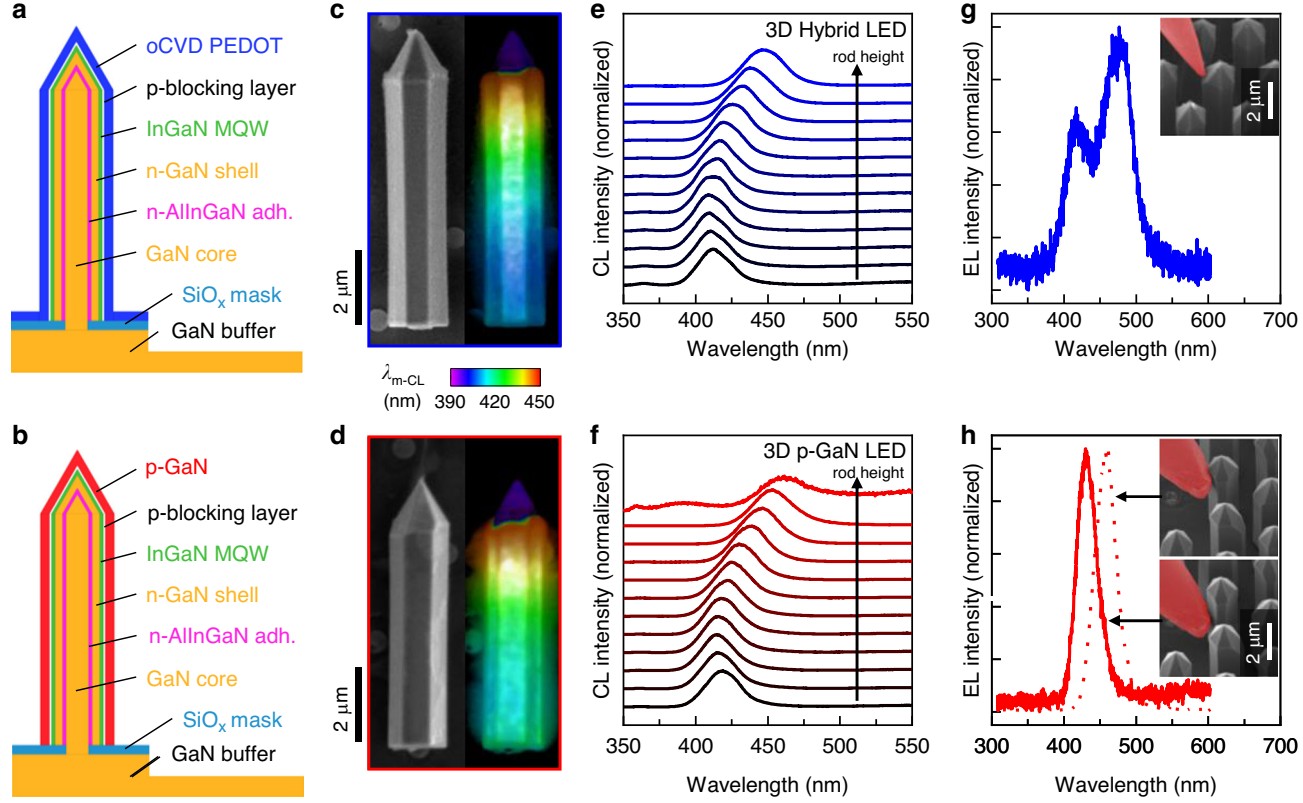

**Fig. 5 3D core-shell hybrid (top panels) and p-GaN (bottom panels) LED structures. a, b** Sketch of the 3D cross-sectional rod structures. The shell around the n-doped core consists of an InGaN-MQW and is capped with a p-doped AlGaN layer. The p-type contact layer is either a conformal PEDOT or a p-GaN layer. **c, d** Scanning electron microscopy (left, scale bar: 2 μm) and hyperspectral cathodoluminescence (CL) maps (right) of individual LED rods indicate a red shift of the InGaN-related luminescence along the rod height which is attributed to a larger concentration of In at the top of the rods. **e, f** CL spectra at different heights of the rods are obtained by shifting a restricted excitation beam area (about $0.5 \times 0.5\,\mu m^2$) with a 0.5 μm step size from the bottom to the top of each rod (indicated by arrows). **g, h** Electroluminescence (EL) spectra recorded at 200 μA. Insets show the position of the p-contact needles highlighted in red (scale bar: 2 μm).

and the oxidant iron(III) chloride (anhydrous $FeCl_3$ purchased from Sigma Aldrich) are provided inside the chamber in the gas phase. The monomer is inserted from a heated monomer jar (140 °C, unless differently stated) and the oxidant is sublimated inside the chamber at temperatures $T_{ox}$ of about 180 °C. After the deposition process, the substrates cool down inside the vacuum chamber. The deposition step is followed by a rinsing step at ambient conditions with methanol or 2-propanol, to remove excess oxidant. Details of the oCVD process can be found in Supplementary Methods.

**Sample fabrication.** All details of the sample fabrication process, especially to the MOVPE GaN growth are presented in Supplementary Methods.

*Sample fabrication of hybrid microrod and hybrid microrod LED structures*: The as-prepared MOVPE-grown GaN microrods were placed in the oCVD vacuum chamber and the oCVD growth process followed the above mentioned oCVD procedure ($T_{substrate} = 125$ °C, $t_{deposition} = 1200$ s). Rinsing was performed using 2-propanol. The hybrid microrod LED structure was prepared accordingly ($T_{substrate} = 150$ °C, $t_{deposition} = 1800$ s) with a 5 min methanol post-deposition rinsing step. Prior to the oCVD process, the MOVPE-grown microrod 4-inch wafer was cut in smaller pieces. To ensure that the microrods do not break during sawing process, the microrods were embedded into an AZ P4620 photoresist matrix that was removed afterward by a 5 min dip in acetone. After the oCVD process, the microrod array is structured into smaller squares with an edge length of about 75 μm using laser ablation. Every small microrod field comprises up to 800 microrods.

*Sample fabrication of hybrid planar samples*: The fabrication of the planar hybrid device begins with a 2″ diameter *c*-plane sapphire substrate covered with a 4 μm thick Si-doped n-GaN layer grown by MOVPE. Covering one half of the sample and applying a lithographically defined mask layer before the oCVD step ($T_{substrate} = 100$ °C, $t_{deposition} = 630$ s) on the other half of the sample produces arrays of PEDOT squares on top of the GaN. The PEDOT has a film thickness of about 110 nm. The squares range between 100 and 400 μm on a side. The oCVD process was followed by a rinsing step with methanol. As contacts, a 100 nm thick gold film was evaporated onto the PEDOT before the removal of the mask. On the PEDOT-free half of the sample, the contacts are similarly deposited after pre-

patterning with a lithographically defined mask. The contacts on n-GaN consist of a stack of 100 nm gold on 30 nm titanium.

*Sample fabrication of hybrid planar LED sample*: The LED substrate was grown via MOVPE. To set the back contact on the n-GaN, first, one part of the sample was etched down using reactive-ion etching whilst the other part of the sample was protected using a mask. The back contact was deposited using thermal evaporation and consists of 300 nm gold on 30 nm titanium. The oCVD process ($T_{substrate} = 180$ °C, $t_{deposition} = 660$ s, $T_{monomer} = 160$ °C) was performed using a shadow mask and the procedure described above. The PEDOT film thickness was estimated to be 640 nm. Finally, the sample was rinsed in methanol and p-contacts were deposited using a shadow mask and thermal evaporation. The contacts consist of a 10 nm palladium layer on which 300 nm gold is deposited with an edge length of 250 μm. Patterning of the PEDOT layer was performed using laser ablation.

**Experimental characterizations.** *I–V* measurements on the planar n-GaN/ PEDOT structures were performed in darkness using a Süss Waferprober system P200 connected to a semiconductor parameter analyzer (Agilent 4155C). Unless differently stated, during a measurement the voltage was swept from −4 to 4 V first and back to −4 V, subsequently. Since the currents for both measurement directions were almost identical for all characteristics investigated here, only the measurements going from positive to negative voltages are presented. For the breakdown measurement, the voltage was swept from 0 V to the maximum reverse voltage and then up again to 0 V. The maximum reverse voltage was increased by steps of 1 V with each measurement. Between each of the measurements down to the maximum reverse voltage, a standard measurement from −4 to 4 V and back was performed in order to investigate whether the characteristics of the diode were changed irreversibly. For the temperature-dependent *I–V* measurements, the sample was heated starting from 273 to 423 K and subsequently cooled down to 273 K in steps of 10 K. For each measurement temperature, the sample was kept 3 min at constant temperature within a tolerance of maximum 1 K before the measurement started.

SEM was performed using a TESCAN Mira 3 GMH field emission scanning electron microscope.

Raman spectra were recorded on a Renishaw inVia Qontor Raman microscope equipped with a ×100 zoom lens and an excitation wavelength of 532 nm. Raman mapping was obtained by scanning the sample with a motorized stage in $xy$-direction at a step size of 1 μm.

$C$–$V$ measurements were performed using an Agilent Semiconductor Device Parameter Analyzer B1500A connected to a LakeShore 8425 Probe Station.

ECV measurements were performed using a Wafer Profiler CVP21.

EL and $I$–$V$ measurements of the hybrid and conventional LED structures were performed using an MPI TS150 Waferprober. The bias voltage is applied using a Keithley 2601B sourcemeter. The EL is detected using a fiber-coupled spectrometer (model AVASPEC-ULS2048LTEC-RS-USB2, Avantes).

EL and CL measurements of microrod LED structures were performed inside a TESCAN Mira 3 GMH field emission scanning electron microscope equipped with a Gatan Mono CL 4 cathodoluminescenc system. An electron energy of 5 keV and a probe current of 100 pA were used for image acquisition. The electrical contacts were established using Kleindiek MM3A micromanipulators with small current measurement kits and tungsten probe tips. The bias current is applied with a Keithley 2636 source measuring unit.

PL of the LEDs was conducted at room temperature under continuous wave excitation (Kimmon, HeCd-Laser, $\lambda_{ex} = 325$ nm). The PL signal of the planar LEDs was obtained by focusing the laser on the surface of the sample under an off-axis angle of ~30° with an energy density of ~1.26 mW mm$^{-2}$ (laser spot ~1.75 mm$^2$, laser power ~2.2 mW) and detected perpendicular to the surface with a fiber-coupled spectrometer (model AVASPEC-ULS2048LTEC-RS-USB2, Avantes). The PL mapping of 3D-LED structures was accomplished by mounting the sample to a motorized xy-stage (Princeton Instruments). The laser was perpendicularly focused on the sample's surface by using a UV-microscope objective (OptoSigma PFL-50-UV/NUV-AG-A) attached to a linear $z$-axis micromanipulator, resulting in an energy density of ~120 W mm$^{-2}$ (laser spot ~1 μm$^2$, laser power ~0.12 mW). The PL signal was collected through a beam splitter and recorded with a spectrometer equipped with a liquid-nitrogen cooled CCD (Princeton Instruments, Acton SP2300, PyLoN). In both cases, planar and 3D LEDs, a long pass filter (AHF, F76-327) was placed in front of the detector to cut-off the light from the laser. The PL map was generated by scanning the sample with a step size of 1 μm using a self-designed program (National Instruments, LabView 2019) to control the xy-stages and CCD.

## Data availability

The data that support the findings of this study are available from the corresponding author upon reasonable request.

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

## Acknowledgements

We thank Klaas Strempel for ECV measurements and Maike Rühmann and Juliane Breitfelder for the processing and lithography of the samples. We acknowledge financial support from "Niedersächsisches Vorab" through "Quantum- and Nano-Metrology (QUANOMET)" initiative within the project NL-3 "Sensors" and NP-3 "Modell-Nanopartikel." We gratefully acknowledge support by the Deutsche Forschungsgemeinschaft (DFG, German Research Foundation) Research Training Group GrK1952/1 "Metrology for Complex Nanosystems" and the Braunschweig International Graduate School of Metrology B-IGSM. This work has been supported by the DFG within the research unit FOR1616 and funded by the DFG under Germany's Excellence Strategy—EXC-2123 QuantumFrontiers—390837967. We acknowledge financial support from the epitaxy competence center ec². M.G., H.v.W., and D.S. acknowledge funding by the European Social Fund within the Young Investigator Group "Oxide Hetero-structures" (SAB 100310460).

## Author contributions

L.K. coordinated the sample preparation, participated at the oCVD growth process, performed electrical measurements, data analysis, and theoretical modeling under the supervision of T.V. Authors S.L. and F.M. performed oCVD on planar LED, GaN microrods, and GaN microrod LEDs under the supervision of T.V. Authors S.L. and S.G. performed EL measurements of the hybrid LED structure with L.K. and F.M. under the supervision of T.V. Authors L.K., D.Splith, and Z.Z. performed the electrical characterization and CV measurements of the hybrid devices under the supervision of H.v.W. and M.G. Author X.W. prepared the oCVD PEDOT layer on planar substrates with L.K. and performed AFM measurements, both under the supervision of K.K.G. Authors J.H., C.M., and I.M.C. performed MOVPE of the microstructured and planar GaN and InGaN/GaN specimen under the supervision of A.W. Authors A.A., T.S., D.Scholz., H.-J.L., and M.S. developed and performed microrod LED growth using MOVPE. S.B. performed laser ablation to structure the planar LED and microrod LED structures. H.S. and F.M. performed EL, CL, and SEM on the microrod LED structures and C.M. did the corresponding data processing of the hyperspectral CL maps, all under the supervision of A.W. Author J.J. analyzed the samples using SEM and F.M. using Raman spectroscopy, both under the supervision of T.V. Author T.V. initiated and supervised this project. L.K., F.M., and T.V. prepared the manuscript. All authors contributed to the manuscript.

## Funding

## Competing interests

The authors declare no competing interests.
