## [Peer Review File · Nature Communications]

Reviewers' Comments:

Reviewer #1:

Remarks to the Author:

The authors report on the potential application of 3D inorganic/organic hybrid optoelectronics with a demonstration of LEDs based on n-GaN/p-oCVD PEDOT. They present conformal coverage of oCVD PEDOT on GaN rods, which may be useful for potential 3D structured optoelectronic applications. Further, the thermal and temporal stability of the hybrid interface with high rectifying performance, which was investigated by planar structures and LED devices, is of importance in hybrid interfaces and their device applications. Overall, the manuscript is well written and covers interesting and impactful findings in the field. Therefore, the reviewer is positive for publication of the manuscript in Nature Communications after successfully addressing the following comments since there are several major issues (mandatory major revision recommendation).

1. Their main goal (bottom line) for the study is to realize 3D structured optoelectronics. To achieve the goal, the authors suggested a hybrid material system of GaN/oCVD PEDOT. Although they had capability to process GaN rods and demonstrated conformal coverage of oCVD PEDOT on GaN rod, the demonstrated devices are all planar, non-3D devices. Since there are already many non-3D hybrid devices available in the literature, the demonstrated non-3D device application does not fully support their goal for the realization of high-performance 3D optoelectronics, although they only provide conformal coating images that are not used for devices.

2. The authors attribute the relatively well performed hybrid LED and its EL property to the improved lateral conductivity (current spreading) of PEDOT. However, this is not solid enough to support their conclusion. Based on their rationale, then, any higher conductivity materials would lead to better performance, which is not true. They should provide more solid evidences to support their conclusion.

3. They should be consistent in use of contact materials. Sometimes they used Au only as metallization for PEDOT for planar devices and they used Pd/Au for PEDOT in LEDs. For n-contact, they sometimes used Ti/Au but sometimes Cr/Au for n-GaN in LEDs. Since there might be some effects on the material/device performance, consistent contact should be made. Further, what is the role of Pd for the p-contact, Pd/Au to PEDOT?

4. As shown in Fig. S4(e), conformal coating, in general, is not favorable for lift-off process even though negative PR is used like in the study (AZ5214). It is clear that the contact shown in the figure is not severely damaged and, therefore, the contact must be enhanced. Further, device characteristics should be affected by the damaged contact. Ways to improve the contact should be discussed to achieve their goal for hybrid optoelectronic devices. These contacts cannot be used for device applications.

5. Although the authors indicated that it is minor, the Au/Ti/n-GaN/Ti/Au shows critically non-ohmic behavior in a voltage regime from ~ -1 to 1, which cannot be neglected. Therefore, their claim that the n-contact is not in charge of the current density is not supported by the data. It seems n-contact is also, at least in part, related to the current density of the device in addition to doping n-GaN concentration.

6. More detailed information such as duration of the annealed sample is to be provided since it is confusing if the annealed data is from annealed samples or from one of the elevated temperature I-V measurements. Although during elevated temperature I-V measurements, some portion of annealing may be progressed, the purpose of measurements and the property of material may be different: for example, annealing generally leads to an irreversible change in material characteristic but short-duration temperature dependent measurements don't.

7. There some typos in the supporting information: in the last paragraph of Section S2, it should

be "Figure S2c) rather than Fig. 3"; doping concentration should be " $2 \times 10^{18} \text{ cm}^{-3}$ vs $2 \times 10^{17} \text{ cm}^{-3}$ " rather than " $2 \times 10^{18} \text{ cm}^{-3}$ vs $2 \times 10^{18} \text{ cm}^{-3}$ ".

8. In Figure 1, the dotted and dashed lines with different colors are not distinguishable and should be fixed.

Reviewer #2:

Remarks to the Author:

The manuscript deals with an interesting and timely topic, namely with the development of a technology for the realization of 3D hybrid inorganic/organic light emitting devices based on the combination of GaN nanowire LED structures and oCVD grown PEDOT layers as an organic, transparent and flexible p-contacting layer.

In the manuscript the conformal deposition of PEDOT on 3D GaN micro LED structures is described and analyzed by SEM and Raman spectroscopy. The electrical properties of a planar PEDOT n-GaN junction were studied by a detailed current-voltage characterization.

Finally, a GaN micro LED device with an InGaN quantum well as active area and an AlGaIn p-blocking layer is contacted with oCVD PEDOT and its performance is compared to that of a "conventional" GaN-micro LED using Mg-doped p-GaN as hole injection layer.

Though addressing a topic of high current interest and high potential impact with regard to the application promised in the title and in the abstract, the presented research work and results do not fulfil the raised expectations.

The analysis of the PEDOT layer deposited on a GaN microrods by SEM impressively confirms the conformal coverage of the rods, which is further confirmed by Raman spectroscopy. However, the PEDOT layers on those planar structures which were used for electrical characterization have not been analyzed. No information about surface roughness and homogeneity (that is of relevance for the breakdown voltage) or structural properties is presented. In addition, as it is essential for modelling that the PEDOT layer can be assumed to have metallic properties, electrical characterization, at least in terms of temperature-dependent sheet conductivity, should be carried out to confirm this assumption. Information on the contact resistance of the Au/PEDOT contacts is also missing.

In order to make the electrical data of the PEDOT layer after aging for 18 months useful, it has to be stated HOW the layer properties have changed in the meantime. Does the layer absorb water? Do structural changes take place? Can oxygen diffuse to the interface?

Regarding electrical characterization, a detailed analysis of planar PEDOT/n-GaN junctions by current-voltage measurements at different temperatures is carried out and the obtained results are explained in a model that takes fluctuations of barrier heights into account. Given the importance of the extracted barrier height for the entire presented work, a second, independent technique for barrier height determination, either capacitance-voltage analysis or bias-dependent photocurrent (which is preferable for pn-type junctions) should be employed. This is of special importance as, although the simplified model of a Schottky junction might be sufficient for quantitative analysis of the current voltage data, it is not clear how the hole-injecting properties of the junction are considered in this model. It has to be stated why modelling by a unipolar junction (not taking minority carriers into account) is a valid approximation, particularly when the junction is not in equilibrium.

Furthermore, the electrical characterization of planar PEDOT/n-GaN junction is only partially relevant for the fabrication and characterization of LED structures, as in those cases the relevant interface is the PEDOT/p-AlGaIn blocking layer. Hence, the part of basic electrical characterization is not really linked to the characterization of the LED structures, as the involved layer stacks are different. For the latter, a comparison of the sheet resistance between the PEDOT layer and the p-GaN layer should be presented (are the thicknesses comparable?). Finally, to prove the potential of PEDOT as a hole injecting layer in GaN-based light emitters, the electrical characteristics of the PEDOT/n-GaN junction should be compared to those of a GaN pn-diode.

More general, although the manuscript contains some interesting aspects, it is mainly of

technological nature and the presented results suffer from missing coherence: In the title, 3D hybrid inorganic/organic devices are mentioned (even if not directly announced). In the electrical characterization PEDOT/n-GaN planar structures are studied. The planar (not 3D) optoelectronic devices that are compared do not contain those interface that were studied before, while the relevant junctions were not studied.

According to the guidelines (" ... in general, to be acceptable, a paper should represent an advance in understanding likely to influence thinking in the field ... ") the different aspects described above converge to the conclusion that the manuscript is not suited for publication in Nature Communications.

Reviewer #3:

None

Reviewer #1:

The authors report on the potential application of 3D inorganic/organic hybrid optoelectronics with a demonstration of LEDs based on n-GaN/p-oCVD PEDOT. They present conformal coverage of oCVD PEDOT on GaN rods, which may be useful for potential 3D structured optoelectronic applications. Further, the thermal and temporal stability of the hybrid interface with high rectifying performance, which was investigated by planar structures and LED devices, is of importance in hybrid interfaces and their device applications. Overall, the manuscript is well written and covers interesting and impactful findings in the field. Therefore, the reviewer is positive for publication of the manuscript in Nature Communications after successfully addressing the following comments since there are several major issues (mandatory major revision recommendation).

1. Their main goal (bottom line) for the study is to realize 3D structured optoelectronics. To achieve the goal, the authors suggested a hybrid material system of GaN/oCVD PEDOT. Although they had capability to process GaN rods and demonstrated conformal coverage of oCVD PEDOT on GaN rod, the demonstrated devices are all planar, non-3D devices. Since there are already many non-3D hybrid devices available in the literature, the demonstrated non-3D device application does not fully support their goal for the realization of high-performance 3D optoelectronics, although they only provide conformal coating images that are not used for devices.

The manuscript so far demonstrated the possibility of hybrid structures for optoelectronic devices. Yet, we did not show light emission from 3D structures. The largest challenge remaining is the contacting of the 3D structures. In response to the reviewers, we performed additional measurements on 3D hybrid n-GaN/InGaN-MQW/p-AlGaIn/PEDOT core-shell microrod structures (multi-quantum-well, MQW). The samples were provided by OSRAM OPTO Semiconductors and the additional polymer shell was deposited via oCVD in Braunschweig. To obtain electroluminescence, the sample was structured into several microrod arrays using laser ablation, placed inside an SEM setup and contacted using two micro-manipulators. One manipulator was placed on a microrod, the other one on the n-GaN layer of the polymer-free substrate next to the microrod array. This area was cleaned and freed of microrods and polymer using laser ablation as proven by SEM, PL and Raman spectroscopy. The figure below, that is also included in the revised manuscript as Fig. 5 shows the sketch of the sample (a), an SEM (b), PL (c) and Raman map of a microrod array, PL (e) of one array and finally electroluminescence (f) of one array/ one microrod. The microrod was, as mentioned above, contacted with a micro-manipulator (inset of figure (f)).

The electroluminescence from these 3D nanoLEDs shows the expected maxima that match the expected wavelength of the InGaN-MQW. This is the first demonstration of a 3D hybrid PEDOT-GaN LED showing the potential of oCVD PEDOT for high-performance 3D optoelectronics.

In the revised manuscript, we included an additional section entitled “3D Microrod hybrid LED-structure”.

2. The authors attribute the relatively well performed hybrid LED and its EL property to the improved lateral conductivity (current spreading) of PEDOT. However, this is not solid enough to support their conclusion. Based on their rationale, then, any higher conductivity materials would lead to better performance, which is not true. They should provide more solid evidences to support their conclusion.

For our hybrid planar LEDs, we observe an enlarged area where the luminescence originates from (revised manuscript Fig. 4c) as compared to a conventional LED (revised manuscript Fig. 4d). We attribute the enlarged light-emitting area to a current spreading by the PEDOT layer because at the comparable purely inorganic LED light emission takes place just beneath and in direct proximity of the contact. As reviewer 1 mentioned, a high lateral conductivity does not necessarily lead to a better performance of an optoelectronic device. It is also necessary, that an efficient hole injection is possible. PEDOT is such a suitable material which exhibits a large work function (estimated for the oCVD PEDOT used of 5.4 eV) [1] and good transparency in the visible spectral range [2,3]. The efficient hole injecting properties can easily be identified by the light emission of our device. In addition, hole injecting properties of PEDOT are discussed in the literature [4–6]. For practical reasons, the material for hole injection should have high transparency in the spectral range where the light emission is taking place if the light extraction is supposed to go through this layer. Low transparency of the hole injection layer leads to a shadowing effect, which lowers the external quantum efficiency of the device due to a reduced light extraction efficiency [7]. In addition, the enlarged current injection area would lead to lower current densities (when total current is kept constant), which finally should lead to better efficiency, when a conventional droop behavior of c-oriented LEDs is assumed.

In the revised manuscript we tried to describe more precisely that the improved current spreading is an additional asset of the PEDOT layer on top of the expected hole-injecting feature. The current spreading explains the enlarged light-emitting area of the device. Changes were made in the introduction of the manuscript as well as in the subsection “Planar hybrid LED-structure”.

3. They should be consistent in use of contact materials. Sometimes they used Au only as metallization for PEDOT for planar devices and they used Pd/Au for PEDOT in LEDs. For n-contact, they sometimes used Ti/Au but sometimes Cr/Au for n-GaN in LEDs. Since there might be some effects on the material/device performance, consistent contact should be made. Further, what is the role of Pd for the p-contact, Pd/Au to PEDOT?

The contacts vary in terms of interlayers as mentioned by the reviewer and summarized in the table below. Therefore, effects on the material/device performance can be expected. Even though consistent contacts are desirable, we deviated from this narrative. In the case of this manuscript, for the p-contact, Au/PEDOT is used for the planar structures and Au/Pd/PEDOT for the hybrid LED. The Pd interlayer of the planar hybrid LED is attributed to the commonly used contact to p-GaN. A later comparison of the hybrid LED to a purely inorganic pn-LED structure is therefore simplified. According to Rass et al. [8], palladium is among those metals with the highest values in work function and therefore most promising for ohmic contacts with p-GaN. Such an ohmic behavior is important for our devices to rule out contributions or a reduction of the efficiency for the performance. Contacting the PEDOT layer is less challenging and Au without an interlayer is sufficient for ohmic contacts as demonstrated in Fig. S6b.

For the n-contact, we are consistent with the planar sample and the planar LED. Both contacts consist of Au/Ti/n-GaN. There is just an inconsistency with the additional hybrid LED structure presented in the SI (formerly section S6 and Fig. S10). The main focus of mentioning the additional sample in the SI is the demonstration of PL and EL measurements and the enlarged area current spreading. As we do a qualitative observation, both purposes are not affected by the Cr interlayer between Au and n-GaN. Therefore, the LED structure in the SI does not diminish the observations presented in the main text but strengthens further the impression of an overall good property of the hybrid device, even with slight modifications. To be more consistent, we have removed the sample containing a Cr interlayer in the revised manuscript. The observations of this sample were included in the revised manuscript with additional PL and EL measurements on the hybrid LED presented (Fig. 4c in the revised manuscript).

We consider the influence of different contacts for the p-contact as negligible for our paper. In both cases for the p-contact we head for ohmic contacts. In addition, as mentioned by the second reviewer, the interface at the LED structure itself is different (PEDOT/n-GaN for the planar samples compared to PEDOT/p-AlGaIn/n-GaN for the LED structure). Thus, as these interfaces are themselves not fully comparable, we preferred to be consistent with common contact stacks used for comparable LED structures with p-GaN instead of PEDOT.

	n-contact	p-contact
Planar n-GaN/p-PEDOT	Au/Ti/n-GaN/...	Au/PEDOT/n-GaN/...
Planar LED structure	Au/Ti/n-GaN/...	Au/Pd/PEDOT/p-AlGaIn/...
Planar LED structure (SI, removed in revised manuscript)	Au/Cr/n-GaN/...	Au/PEDOT/n-GaN/...

In order to be more consistent in the manuscript, we removed the sample presented in the SI and focused on the parameters of the hybrid structure presented in Fig. 4 with the help of additional

measurements (PL and EL).

4. As shown in Fig. S4(e), conformal coating, in general, is not favorable for lift-off process even though negative PR is used like in the study (AZ5214). It is clear that the contact shown in the figure is not severely damaged and, therefore, the contact must be enhanced. Further, device characteristics should be affected by the damaged contact. Ways to improve the contact should be discussed to achieve their goal for hybrid optoelectronic devices. These contacts cannot be used for device applications.

We completely agree with the reviewer, the contacts shown are not sufficient for commercial device applications. For the study, we carefully selected contacts with good quality. We did not use the contact shown in the figure mentioned (Fig. S4e). The contact shown in Fig. S4e was chosen to demonstrate the polymer layer beneath the Au contact layer.

Contact processing is a crucial factor for device fabrication. The n-contacts of our structures consist of Au/Ti which is well established for n-GaN contacts. For the p-contacts, we can picture several optimization steps. These steps comprise annealing of the Au contacts in an inert atmosphere or low temperature deposition of the Au contact. Indeed, the structuring of PEDOT can be challenging and might negatively impact the process yield (number of suitable pn-junctions on the wafer). In the present study, we attempted the lift-off approach and obtained several desired square-shaped p-contacts of PEDOT/Au (cf. Fig 2b) with a high IV reproducibility.

In order to analyze the optical properties of the hybrid LED structures, it is crucial that the ohmic contact covers a smaller area than the PEDOT. During our study, we noticed that conventional lithography is less suitable after depositing the oCVD PEDOT thin film (e.g. structural changes of the PEDOT polymer due to the chemicals used during lithography). For this reason, it was crucial, to develop a patterning technique of the PEDOT layer that works without lithography. One attempt is the use of laser ablation which we used for structuring the polymer layers for the hybrid LED devices. At present, laser ablation is the best choice for patterning. This improves the contacts to the necessary extent for future device applications.

The advantages of laser ablation can be seen in the new included Fig. 5 in the revised manuscript (Figure presented already in the answer to question 1). As can be seen with Fig 5(b-d) with the help of SEM as well as PL and Raman spectroscopy, laser ablation removes GaN and PEDOT reliably without contamination of any surface through photoresists or shadow masks. The strength is also demonstrated in Fig. 4c and 4d (revised manuscript) for planar polymer layers. For the hybrid planar LED, the polymer layer was patterned using laser ablation.

Therefore, for hybrid optoelectronic devices, we recommend to pattern polymer layers with the help of laser ablation or – not presented in this manuscript, but also tested in our laboratories – reactive ion etching. The Au contacts on the polymer layer can finally be deposited using shadow masks and thermal evaporation.

We included in the revised manuscript (Supplementary Section S3 “Additional information about contacts and breakdown voltages”) a shortened discussion of the above-mentioned optimization possibilities for the contacts on the PEDOT layer.

5. Although the authors indicated that it is minor, the Au/Ti/n-GaN/Ti/Au shows critically non-ohmic behavior in a voltage regime from ~ -1 to 1, which cannot be neglected. Therefore, their claim that the n-contact is not in charge of the current density is not supported by the data. It seems n-contact is also, at least in part, related to the current density of the device in addition to doping n-GaN concentration.

The contribution of the contacts on the diode characteristics is negligible as the magnitude of the current density of the contacts is much higher than those recorded from the diode (Fig. 2C). Differently stated, as long as the current from the diode is not in the order of the current from the contact, the deviations from perfect ohmic behavior are not relevant.

In the manuscript, we added the above-mentioned reason for the disregard of the deviation of perfect ohmic behavior and we changed the phrase “a minor s-shape behavior” to “an s-shape behavior” in the SI (Subtitle of Fig. S6 and Section S3).

6. More detailed information such as duration of the annealed sample is to be provided since it is confusing if the annealed data is from annealed samples or from one of the elevated temperature I-V measurements. Although during elevated temperature I-V measurements, some portion of annealing may be progressed, the purpose of measurements and the property of material may be different: for example, annealing generally leads to an irreversible change in material characteristic but short-duration temperature dependent measurements don't.

“Annealing” in the submitted manuscript is always referring to the heating process in connection with the IV temperature series. The temperature series itself took 200 minutes starting with 20°C, heating to 150°C and cooling back to 20°C again. Out of these 200 minutes, 115 minutes the sample was above 100°C. We don't expect significant changes in the polymer layer due to the heating process as the deposition of the polymer layer was performed at comparable temperatures (100°C). This expectation is supported by minor changes in the electrical characteristics after heating.

We changed the word “annealed” in the revised version of the manuscript to “after heating” (Fig. 2C, Fig. S2 and in section S2).

7. There some typos in the supporting information: in the last paragraph of Section S2, it should be “Figure S2c) rather than Fig. 3”; doping concentration should be “ $2 \times 10^{18} \text{ cm}^{-3}$ vs $2 \times 10^{17} \text{ cm}^{-3}$ ” rather than “ $2 \times 10^{18} \text{ cm}^{-3}$ vs $2 \times 10^{18} \text{ cm}^{-3}$ ”.

The typos in Section S2 were corrected.

8. In Figure 1, the dotted and dashed lines with different colors are not distinguishable and should be fixed.

In Fig. 1 we increased the linewidth of the dashed line for the PEDOT line and thinned and dotted the lines for PEDOT. In addition, we labeled the lines on the top of the graph.

Reviewer #2 (Remarks to the Author):

The manuscript deals with an interesting and timely topic, namely with the development of a technology for the realization of 3D hybrid inorganic/organic light emitting devices based on the combination of GaN nanowire LED structures and oCVD grown PEDOT layers as an organic, transparent and flexible p-contacting layer.

In the manuscript the conformal deposition of PEDOT on 3D GaN micro LED structures is described and analyzed by SEM and Raman spectroscopy. The electrical properties of a planar PEDOT n-GaN junction were studied by a detailed current-voltage characterization.

Finally, a GaN micro LED device with an InGaN quantum well as active area and an AlGaIn p-blocking layer is contacted with oCVD PEDOT and its performance is compared to that of a “conventional” GaN-micro LED using Mg-doped p-GaN as hole injection layer.

1) Though addressing a topic of high current interest and high potential impact with regard to the application promised in the title and in the abstract, the presented research work and results do not fulfil the raised expectations.

We have performed additional experiments to address this issue. In the revised manuscript, we present an additional comprehensive analysis of hybrid 3D nanoLEDs. Details have been discussed in the answer to the first question of reviewer 1.

2) The analysis of the PEDOT layer deposited on a GaN microrods by SEM impressively confirms the conformal coverage of the rods, which is further confirmed by Raman spectroscopy. However, the PEDOT layers on those planar structures which were used for electrical characterization have not been analyzed. No information about surface roughness and homogeneity (that is of relevance for the breakdown voltage) or structural properties is presented.

PEDOT layers deposited with oCVD are regularly analyzed using four-point probe and AFM to determine the roughness, sheet resistance and lateral conductivity [3,9,10]. Typical values for the rms roughness are 0.7 nm [9], 3.8 nm [11] and 2,85 nm [10].

The figure below shows a typical AFM image of a PEDOT layer deposited at comparable conditions like the PEDOT film presented (same reactor, substrate temperature during deposition of 150°C). The estimated surface roughness is 2 nm.

Phenomenologically, the magnitude of the breakdown voltage has been proven to be reasonable as it was reproduced on the sample reported and its equivalent structure. The importance of the estimated breakdown voltage is the electrical stability of the hybrid junction.

In the revised manuscript we included the above-mentioned estimation of the roughness of the PEDOT layer and included the presented AFM image in the supplementary figure S4f. Furthermore, we weakened the description of our observations concerning the breakthrough voltage.

3) In addition, as it is essential for modelling that the PEDOT layer can be assumed to have metallic properties, electrical characterization, at least in terms of temperature-dependent sheet conductivity, should be carried out to confirm this assumption. Information on the contact resistance of the Au/PEDOT contacts is also missing.

In general, the metallic-like conductivity of (oCVD) PEDOT with high conductivity has been established in references [10,12,13].

Sheet resistance measurements and the determination of the conductivity of the oCVD polymer layer are regularly measured and reported in the literature. Usually, the conductivity of the polymer layers deposited using oCVD at the growth conditions described is in the order of several 100 S/cm up to several 1000 S/cm [3,10,11,14]. The samples described for this paper consist of a patterned polymer layer. The individual polymer patches are completely covered with contacts, therefore, it is not possible to measure their sheet conductivity.

We note that the in-plane and the out-of-plane conductivities of PEDOT show a large anisotropy. Depending on the orientation of the polymer layer (“face-on” or “edge-on”) the anisotropy ratio has a value in the order of 10^4 to 10^6 [10]. Therefore, the in-plane conductivity is much larger than the out-of-plane conductivity. Further studies of the out-of-plane conductivities of PEDOT:PSS were performed by Ruit et al. [15] and Fuji and co-workers [16]. Whilst for the later discussion of the current spreading in the planar hybrid LEDs the lateral conductivity is important, for the planar hybrid n-GaN/PEDOT structures the out-of-plane conductivity is relevant for the modeling of IV measurements. The out-of-plane conductivities can't be obtained from temperature-dependent sheet conductivity measurements.

The quasi-metallic or metallic state of PEDOT and its derivatives is commonly accepted in literature. It is usually attributed to large doping and band structure calculations yield a metallic behavior [17–21]. We followed this assumption and the assumption is supported by our results. However, one has to keep in mind, that, as stated before, most analyses are taken for in-plane PEDOT behavior.

For the Au/PEDOT contacts, we showed in the SI the ohmic IV characteristics of Au on PEDOT on a similar sample (Fig. S6b). The structure we used for the submitted manuscript didn't allow for purely Au/PEDOT measurements due to the sample design. Such an analysis would only be possible before patterning the PEDOT layer.

4) In order to make the electrical data of the PEDOT layer after aging for 18 months useful, it has to be stated HOW the layer properties have changed in the meantime. Does the layer absorb water? Do structural changes take place? Can oxygen diffuse to the interface?

Despite previous degradation studies of oCVD PEDOT at ambient conditions over time, our experimental results proved that there are negligible aging effects after 18 weeks at ambient conditions for the device. Additional measurements revealed even after 113 weeks no clear degradation. This is unexpected as Chen et al. [22] reported a loss of about half of the initial conductivity of PEDOT within 115 hours if stored at 30°C in air. The degradation is accelerated if the storage temperature is raised to 50°C (61 hours) and 100°C (1.5 hours). The degradation of the conductivity is attributed to two major effects. On the one hand, H₂O or O₂ diffuses into the film. This is especially severe if there are still hygroscopic excess oxidant within or on top of the PEDOT layer. We tried to remove as much as possible of the excess oxidant using a post-deposition 10 minutes methanol rinsing step for the sample. On the other hand, the dopant diffuses out of the film. Encapsulation of the polymer film reduces the degradation of the polymer film in terms of its conductivity. We assume, that the Au contact on top of the PEDOT layer acts as an encapsulation layer, therefore, preventing both, diffusion into the film as well as diffusion out of the film. However, the degradation studies performed by Chen et al. focused on the lateral in-plane conductivity of the polymer film. Depending on the orientation of the polymer chains, the degradation of the conductivity in transversal (in-device) direction has not been analyzed yet. Presumably, the degradation processes would be the same. We could not identify any changes of the sample in the meantime.

We note: All these considerations are made about the lateral conductivity of the polymer layer. What we discuss in our paper, is the transversal conductivity that seems, according to the obtained data, not to be significantly degraded.

We included a shortened version of the above-mentioned considerations into the manuscript (Section “Electrical characterization of planar n-GaN/PEDOT structures”).

5) Regarding electrical characterization, a detailed analysis of planar PEDOT/n-GaN junctions by current-voltage measurements at different temperatures is carried out and the obtained results are explained in a model that takes fluctuations of barrier heights into account. Given the importance of the extracted barrier height for the entire presented work, a second, independent technique for barrier height determination, either capacitance-voltage analysis or bias-dependent photocurrent (which is preferable for pn-type junctions) should be employed. This is of special importance as, although the simplified model of a Schottky junction might be sufficient for quantitative analysis of the current voltage data, it is not clear how the hole-injecting properties of the junction are

considered in this model. It has to be stated why modelling by a unipolar junction (not taking minority carriers into account) is a valid approximation, particularly when the junction is not in equilibrium.

Capacitance-voltage (CV) and quasi-static CV (QSCV) analysis were performed and yielded equivalent electronic barrier heights compared to the mean barrier height as obtained via temperature-dependent IV measurements. The values are in good agreement for the sample reported (IV: 1.42 ± 0.01 eV; CV: 1.39 ± 0.01 eV; QSCV: 1.32 ± 0.01 eV) and its equivalent sample (IV: 1.49 ± 0.01 eV; CV: 1.35 ± 0.01 eV, QSCV: 1.31 ± 0.01 eV).

Concerning the hole-injecting properties of the junction, our results show that in the case of the description of the hybrid PEDOT/n-GaN junction the current can be well described by electronic transport without considering hole injection. We also wish to emphasize that no light emission was observed for all PEDOT/n-GaN layers studied in this work for the whole range of applied biases up to 8 V which strengthens our claim.

For the case of the hybrid LED, however, the specific structure leads to hole injection into the MQW as evidenced by the light emission. The introduction of a 30 nm p-AlGa_N blocking layer can be understood as the introduction of an intervening dielectric layer. We assume that due to the layer, a sufficient potential energy difference at the electronic interface can be maintained that allows for tunneling of holes from the PEDOT into the MQW. Studies on the introduction of an intervening dielectric layer were performed by Zimmler et al. [23]. For the material system n-ZnO/p-Silicon they observed that only small changes like the introduction of a layer of few nm SiO₂ can substantially change the electronic structure of the junctions and the electronic conduction mechanism.

In the revised manuscript, we included the obtained results for the CV analysis. Furthermore, as these measurements were measured 113 weeks after the initial measurements, we replaced in Fig. 2c the IV characteristics “after 18 weeks” with the ones obtained “after 113 weeks”.

6) Furthermore, the electrical characterization of planar PEDOT/n-GaN junction is only partially relevant for the fabrication and characterization of LED structures, as in those cases the relevant interface is the PEDOT/p-AlGa_N blocking layer. Hence, the part of basic electrical characterization is not really linked to the characterization of the LED structures, as the involved layer stacks are different. For the latter, a comparison of the sheet resistance between the PEDOT layer and the p-GaN layer should be presented (are the thicknesses comparable?).

We fully agree with the reviewer. It is our intention to divide the manuscripts into several parts that address different levels of complexity regarding the analyzed hybrid structure. The first part addresses the PEDOT/n-GaN junctions, and provides important information that lay the foundations for the study of the complex hybrid LED structures: We are able to obtain stable and reproducible hybrid junctions that exhibit electrical characteristics which can be readily explained by applying the Schottky model. The electrical properties of the hybrid interface are not largely influenced by defect states.

In the second part, we study the optoelectronic properties of the full hybrid LED structures and use both the electronic and optical properties to demonstrate hole injection from the PEDOT into the InGa_N quantum wells. A direct comparison of the IV characteristics of a standard Ga_N-based LED and the hybrid LED is now given in Fig. 4 (g and h) of the revised manuscript (also included for the answer on question 7 of the reviewer). In this figure, the formation of an electronic structure after PEDOT deposition can be seen that closely resembles that of the standard Ga_N LED. The rectification ratio of both structures is in the order of 10^4 at ± 4 V. In reverse direction, the conventional LED yields a faster increase of the leakage current at voltages between 0 and -2 V. The series resistance dominates the conventional LED structure at about 2 V, the hybrid structure is limited by the series

resistance above 3.5 V. From the IV-characteristics, we estimate comparable series resistances of the two devices, namely $(170 \pm 79) \Omega$ and $(84 \pm 7) \Omega$ for the hybrid and conventional LED structure, respectively (differing by not more than a factor two). This estimation strengthens the point that the PEDOT layer provides improved lateral conductivity compared to the well-established p-GaN without a significant increase of the series resistance

Furthermore, the observation of electroluminescence from the InGaN QW is a clear demonstration of hole injection into the MQW structure. We, therefore, believe that our results, taken together, provide a quite clear and thorough picture of the electronic properties, starting from the simple case of the PEDOT/n-GaN junction all the way to the complex hybrid LED structure.

As the reviewer points out, comparing the sheet resistance of p-GaN and PEDOT is especially helpful if the layer thicknesses are the same. The PEDOT layer thickness on the hybrid planar LED structure amounts to around $(640 \pm 37) \text{ nm}$ and is therefore 4 times as thick as the stack of two p-GaN layers which sums up to around 120 nm. Therefore, the value one wants to compare is the resistivity or the conductivity of the films. The sheet resistance of PEDOT was determined on equivalent samples, where the PEDOT layer was deposited on glass. The glass substrate was placed in the deposition chamber at the very same run as the planar LED structure. The sheet resistance is $(205 \pm 24) \text{ Ohm}/\square$ for the PEDOT layer (lateral conductivity: $(76 \pm 15) \text{ S/cm}$). As for the p-GaN layer, the determination of the sheet resistance is technologically challenging because of the necessary p-GaN layer growth on sapphire or any insulating substrate. Therefore, we refer for the values of p-GaN to literature values reported by Morkoc et al. [24]. Morkoc reported that hole mobilities for p-GaN are in the order of $10\text{-}20 \text{ cm}^2/\text{Vs}$ and hole concentrations can be up to $1 \cdot 10^{18} /\text{cm}^3$. Together one calculates conductivities up to 3.2 S/cm .

Hall measurements at our institute for older comparable structures where the sample consists of a thin p-AlGaIn layer and 600 nm p-GaN on an n-GaN buffer yielded mobilities of $8.8 \text{ cm}^2/\text{Vs}$ and hole concentrations of $2.5 \cdot 10^{17} /\text{cm}^3$. With this, we obtained a conductivity of 0.35 S/cm . Assuming very good values for mobility ($80 \text{ cm}^2/\text{Vs}$) and hole concentration ($1 \cdot 10^{18} /\text{cm}^3$), we would calculate a conductivity of about 13 S/cm which is still less than a quarter of the value measured for PEDOT.

We included into the manuscript (Section "Planar hybrid LED-structure") the values of the lateral conductivity for the PEDOT layer and an estimation of the conductivity of the p-GaN layer

7) Finally, to prove the potential of PEDOT as a hole injecting layer in GaN-based light emitters, the electrical characteristics of the PEDOT/n-GaN junction should be compared to those of a GaN pn-diode.

Again, we fully agree with the reviewer, and in response to the report, we have performed a large set of new measurements and amended Fig.4 with further results as requested by the reviewer. In particular, we now give a direct comparison of the IV characteristics of a standard GaN-based LED (with p-GaN) (see Fig. 4h) and the hybrid LED (see Fig. 4g) in which the p-GaN has been replaced by the PEDOT layer. As discussed in the response to question 6, the electrical properties for hybrid and the conventional inorganic LED resemble each other closely.

The light emission demonstrates hole injection from the PEDOT/p-AlGaIn layers into the MQW region of the LED. As can be seen from the PL and EL spectra of the hybrid (Fig. 4e) and conventional (Fig 4f) LED, the overall luminescent properties remain equivalent.

In the revised manuscript, we included the extended Fig. 4 and discussed the differences of the hybrid with the conventional LED in the section "Planar hybrid LED-structure".

8) More general, although the manuscript contains some interesting aspects, it is mainly of technological nature and the presented results suffer from missing coherence: In the title, 3D hybrid inorganic/organic devices are mentioned (even if not directly announced). In the electrical characterization PEDOT/n-GaN planar structures are studied. The planar (not 3D) optoelectronic devices that are compared do not contain those interface that were studied before, while the relevant junctions were not studied. According to the guidelines (“... in general, to be acceptable, a paper should represent an advance in understanding likely to influence thinking in the field ...”) the different aspects described above converge to the conclusion that the manuscript is not suited for publication in Nature Communications.

We have taken the criticism of the reviewer very seriously and have undertaken substantial effort in increasing the coherence of the manuscript and filling the logical gaps. In particular, we now present new studies of the optoelectronic characteristics of hybrid 3D PEDOT/GaN microrod LEDs and compare the optical and electronic characteristics with state-of-the-art conventional inorganic GaN-based LEDs. We have already outlined the details of these new results in the answers to the previous questions in detail. We would like to stress at this point, that our work contains many aspects that are directly related to fundamental science (i.e. the electronic properties of the hybrid PEDOT/GaN interface with the IV and (new) CV analysis), technology (i.e. oCVD) as well as applications (i.e. LED-structure).

To our understanding, the manuscript now has a fully coherent story, and in particular addresses all the questions, concerns, and requests that were raised by the reviewers in detail. In addition, the problem of making contacts to 3D structures with high aspect ratios is getting increasingly important due to the many research activities towards micro- and nanoLEDs, which implicitly will have very large aspect ratios. This brings in important relevance of our approach to future micro- and nanoLED technology, e.g. in the field of micro-displays for augmented reality, quantum computation or neuromorphic networks based on nanoLED arrays. In all these cases, 3D structures with very high aspect ratios have to be put into contact with their environment. We feel that our work has the

necessary scientific background as well as technological relevance to be interesting for a large scientific community.

We would like to explicitly thank the reviewers for providing very critical, but constructive reviews that greatly helped us to substantially improve the quality of the manuscript, including a lot of additional experimental work. We, therefore, believe that the manuscript will be suited for publication in Nature Communications in the present form.

Bibliography

- [1] Im S G and Gleason K K 2007 Systematic control of the electrical conductivity of poly(3,4-ethylenedioxythiophene) via oxidative chemical vapor deposition *Macromolecules* **40** 6552–6
- [2] Gharahcheshmeh M H, Tavakoli M M, Gleason E F, Robinson M T, Kong J and Gleason K K 2019 Tuning, optimization, and perovskite solar cell device integration of ultrathin poly(3,4-ethylene dioxythiophene) films via a single-step all-dry process *Sci. Adv.* **5** 1–13
- [3] Howden R M, McVay E D and Gleason K K 2013 OCVD poly(3,4-ethylenedioxythiophene) conductivity and lifetime enhancement via acid rinse dopant exchange *J. Mater. Chem. A* **1** 1334–40
- [4] Kim Y H, Sachse C, Machala M L, May C, Müller-meskamp L and Leo K 2011 Highly Conductive PEDOT : PSS Electrode with Optimized Solvent and Thermal Post-Treatment for ITO-Free Organic Solar Cells *Adv. Funct. Mater.* **21** 1076–81
- [5] Riedel B, Shen Y, Hauss J, Aichholz M, Tang X, Lemmer U and Gerken M 2011 Tailored highly transparent composite hole-injection layer consisting of pedot:PSS and SiO₂ nanoparticles for efficient polymer light-emitting diodes *Adv. Mater.* **23** 740–5
- [6] Groenendaal L, Jonas F, Freitag D, Pielartzik H and Reynolds J R 2000 Poly(3,4-ethylenedioxythiophene) and its derivatives: past, present, and future *Adv. Mater.* **12** 481–94
- [7] Kneissl M, Seong T Y, Han J and Amano H 2019 The emergence and prospects of deep-ultraviolet light-emitting diode technologies *Nat. Photonics* **13** 233–44
- [8] Kneissl M and Rass J 2016 *III-Nitride Ultraviolet Emitters - Technology and Applications* ed M Kneissl and J Rass (Heidelberg: Springer)
- [9] Park H, Howden R M, Barr M C, Bulović V, Gleason K and Kong J 2012 Organic solar cells with graphene electrodes and vapor printed poly(3,4-ethylenedioxythiophene) as the hole transporting layers *ACS Nano* **6** 6370–7
- [10] Wang X, Zhang X, Sun L, Lee D, Lee S, Wang M, Zhao J, Shao-Horn Y, Dincă M, Palacios T and Gleason K K 2018 High electrical conductivity and carrier mobility in oCVD PEDOT thin films by engineered crystallization and acid treatment *Sci. Adv.* **4** 1–9
- [11] Lee S, Paine D C and Gleason K K 2014 Heavily doped poly(3,4-ethylenedioxythiophene) thin films with high carrier mobility deposited using oxidative CVD: Conductivity stability and carrier transport *Adv. Funct. Mater.* **24** 7187–96
- [12] Farka D, Coskun H, Gasiorowski J, Cobet C, Hingerl K, Uiberlacker L M, Hild S, Greunz T, Stifter D, Sariciftci N S, Menon R, Schoefberger W, Mardare C C, Hassel A W, Schwarzinger C, Scharber M C and Stadler P 2017 Anderson-Localization and the Mott–Ioffe–Regel Limit in

Glassy-Metallic PEDOT *Adv. Electron. Mater.* **3** 1–8

- [13] Farka D, Jones A O F, Menon R, Serdar N and Stadler P 2018 Metallic conductivity beyond the Mott minimum in PEDOT : Sulphate at low temperatures *Synth. Met.* **240** 59–66
- [14] Ugur A, Katmis F, Li M, Wu L, Zhu Y, Varanasi K K and Gleason K K 2015 Low-Dimensional Conduction Mechanisms in Highly Conductive and Transparent Conjugated Polymers *Adv. Mater.* **27** 4604–10
- [15] Van De Ruit K, Katsouras I, Bollen D, Van Mol T, Janssen R A J, De Leeuw D M and Kemerink M 2013 The curious out-of-plane conductivity of PEDOT:PSS *Adv. Funct. Mater.* **23** 5787–93
- [16] Fujii S, Suzuki Y, Kawamata J and Tsunashima R 2015 Large in-plane/out-of-plane anisotropic conduction in PEDOT-based hybrid films: Lamellar assemblies structured by mono-layered nanosheets *J. Mater. Chem. C* **3** 7153–8
- [17] Kim E G and Brédas J L 2008 Electronic evolution of poly(3,4-ethylenedioxythiophene) (PEDOT): From the isolated chain to the pristine and heavily doped crystals *J. Am. Chem. Soc.* **130** 16880–9
- [18] Bubnova O, Khan Z U, Wang H, Braun S, Evans D R, Fabretto M, Hojati-Talemi P, Dagnelund D, Arlin J B, Geerts Y H, Desbief S, Breiby D W, Andreasen J W, Lazzaroni R, Chen W M, Zozoulenko I, Fahlman M, Murphy P J, Berggren M and Crispin X 2014 Semi-metallic polymers *Nat. Mater.* **13** 190–4
- [19] Franco-Gonzalez J F and Zozoulenko I V. 2017 Molecular Dynamics Study of Morphology of Doped PEDOT: From Solution to Dry Phase *J. Phys. Chem. B* **121** 4299–307
- [20] Rudd S, Franco-Gonzalez J F, Kumar Singh S, Ullah Khan Z, Crispin X, Andreasen J W, Zozoulenko I and Evans D 2018 Charge transport and structure in semimetallic polymers *J. Polym. Sci. Part B Polym. Phys.* **56** 97–104
- [21] Shi W, Zhao T, Xi J, Wang D and Shuai Z 2015 Unravelling Doping Effects on PEDOT at the Molecular Level: From Geometry to Thermoelectric Transport Properties *J. Am. Chem. Soc.* **137** 12929–38
- [22] Chen N, Kovacic P, Howden R M, Wang X, Lee S and Gleason K K 2015 Low substrate temperature encapsulation for flexible electrodes and organic photovoltaics *Adv. Energy Mater.* **5** 1–7
- [23] Zimmler M A, Stichtenoth D, Ronning C, Yi W, Narayanamurti V, Voss T and Capasso F 2008 Scalable fabrication of nanowire photonic and electronic circuits using spin-on glass *Nano Lett.* **8** 1695–9
- [24] Morkoç H 2008 *Nitride Semiconductor Devices : Principles and Simulation Properties of Group-IV , III-V and II-VI Semiconductors Nitride Semiconductors*

Reviewers' Comments:

Reviewer #1:

Remarks to the Author:

Overall, the manuscript has been much enhanced by addressing the raised claims and suggestions, particularly by the demonstration of 3D hybrid inorganic/organic LEDs. However, the current version lacks the rationale for use of the suggested hybrid structure, compared to inorganic (non-hybrid) 3D LEDs. A clear way to resolve the issue and provide the rationale is to compare the performance of the demonstrated 3D hybrid LED to those of inorganic (non-hybrid) LEDs that employ p-GaN as p-contact (although the conductivity of p-GaN is low).

Therefore, the reviewer suggests the authors need to fabricate the identical 3D LEDs with p-GaN (instead of PEDOT), which can be used as control-sample to prove the role and effect of organic p-contact and provide rationale for the suggested hybrid LEDs.

Reviewer #2:

Remarks to the Author:

The authors have carefully considered all of the reviewers' comments. In most cases they have revised and significantly improved the manuscript. The direct technical issues have been addressed and almost all of the weak points have been eliminated and the respective parts are solidified.

The presented work presents significant advancement in the field of hybrid LED device technology that will impact and initiate further work.

Reviewer #1:

Overall, the manuscript has been much enhanced by addressing the raised claims and suggestions, particularly by the demonstration of 3D hybrid inorganic/organic LEDs. However, the current version lacks the rationale for use of the suggested hybrid structure, compared to inorganic (non-hybrid) 3D LEDs. A clear way to resolve the issue and provide the rationale is to compare the performance of the demonstrated 3D hybrid LED to those of inorganic (non-hybrid) LEDs that employ p-GaN as p-contact (although the conductivity of p-GaN is low).

Therefore, the reviewer suggests the authors need to fabricate the identical 3D LEDs with p-GaN (instead of PEDOT), which can be used as control-sample to prove the role and effect of organic p-contact and provide rationale for the suggested hybrid LEDs.

We would like to thank the reviewer once more for the critical, constructive and helpful feedback to our submission. In response to the reviewer's requests, we performed additional measurements on a nearly identical, purely inorganic 3D LED structure with p-GaN instead of PEDOT as a p-contact. These measurements were taken as a reference to judge on the performance of the hybrid LED with respect to conventional inorganic GaN-based LEDs.

The measurements were kept analogously to the previous measurements of the hybrid microrod core-shell LEDs. In particular, the measurements comprised electroluminescence (EL) and photoluminescence (PL). In addition, we now included cathodoluminescence (CL) measurements for both, hybrid and purely inorganic core-shell microrods. Results of those measurements are shown in Fig. 5 (see below) which has been rearranged in order to provide a clear comparison between the optical performance of a hybrid (top panels) and a purely inorganic (bottom panels) 3D LED. In this process, some results that were previously included in Fig. 5 have been moved to the supplementary information (figures S10-S12).

The new measurements clarify the role of the organic p-contact as compared to the purely inorganic p-GaN shell: PEDOT features a substantially improved current spreading which allows to contact a larger area of the microrod at a time leading to a broader optical spectrum of the electroluminescence. The discussion has the following rationale. In the new Figure 5, we show the results of cathodoluminescence investigations on single 3D LEDs that demonstrate a red shift of the optical emission when moving the excitation spot from bottom to top. This phenomenon has additionally been proven by PL measurements (figure S12) and has previously been observed for such LEDs [1,2]. The red shift is a consequence of a gradient of the indium incorporation in the multi-quantum well on the sidewalls (lower indium content at the bottom, higher indium content at the top).

Figure 5. 3D core-shell hybrid (top panels) and p-GaN (bottom panels) LED structures.

Electroluminescence of the purely inorganic 3D LED (i.e., with p-GaN) shows a distinct red shift of the emission line when the p-side contact needle is moved from bottom to top. This is in contrast to the substantially broader EL spectrum of the hybrid LED. The increased electroluminescence line width of the hybrid LED is a clear indication that in this case the improved current spreading of the PEDOT layer leads to simultaneous electroluminescence from the top AND the bottom parts of the 3D LED. This clearly demonstrates the improved optical performance of the hybrid LED with respect to the inorganic reference LED.

In the new manuscript, we added the measurements concerning the inorganic reference microrod LED structure in the new arranged figure 5 and discussed the results in the revised section “3D Microrod hybrid LED structure”. The previously shown SEM-, Raman- and PL-maps, that highlight the feasibility of laser ablation for structuring the microrod arrays, were moved into the SI as the new figure S10. Furthermore, we divided the (former) figure S10 from the previously submitted manuscript into two separate figures (S11 – diameter evaluation and S12 - PL) and added the corresponding measurements of the non-hybrid LEDs.

Reviewer #2:

The authors have carefully considered all of the reviewers' comments. In most cases they have revised and significantly improved the manuscript. The direct technical issues have been addressed and almost all of the weak points have been eliminated and the respective parts are solidified.

The presented work presents significant advancement in the field of hybrid LED device technology that will impact and initiate further work.

We thank both reviewers once again for the valuable and constructive feedback which helped and motivated us to improve the manuscript to a large extent!

Bibliography

- [1] Mounir C, Schimpke T, Rossbach G, Avramescu A, Strassburg M and Schwarz U T 2017 Polarization-resolved micro-photoluminescence investigation of InGaN/GaN core-shell microrods *J. Appl. Phys.* **121**
- [2] Müller M, Veit P, Krause F F, Schimpke T, Metzner S, Bertram F, Mehtens T, Müller-Caspary K, Avramescu A, Strassburg M, Rosenauer A and Christen J 2016 Nanoscopic Insights into InGaN/GaN Core-Shell Nanorods: Structure, Composition, and Luminescence *Nano Lett.* **16** 5340–6

Reviewers' Comments:

Reviewer #1:

Remarks to the Author:

The reviewer believes that the authors have well addressed the issues raised by the reviewers. Overall, the manuscript has been much improved compared to the initial version and the current version would be suitable for publication in Nature Communications.

Reviewer #1:

The reviewer believes that the authors have well addressed the issues raised by the reviewers. Overall, the manuscript has been much improved compared to the initial version and the current version would be suitable for publication in Nature Communications.

The authors thank both reviewers for the constructive support throughout the whole review process!